# Mid-gestational cell-type-specific transcriptomic signatures in the prefrontal and superior temporal cortex in Down syndrome

Rui-Ze Niu[1,2,7], Lu-Lu Xue[1,3,7], Xiao-He Tian [1,7], Li-Ren Huangfu[4], Li Chen[3], Chen-Yang Zhai[4], Shi-Feng Wang[1], Yang-Yang Zhao[4], Zong-Jin Gan[1], Hao-Yue Qin[1], Ting-Hua Wang[1] ✉, Cheng Liu[5] ✉ & Liu-Lin Xiong [1,6] ✉

The cellular and molecular mechanisms underlying cortical alterations during early fetal development in Down syndrome (DS) remain largely unexplored. Here, we perform single-nucleus RNA sequencing (snRNA-seq) analysis on mid-gestational DS and control brain samples, including prefrontal cortex (PFC) and superior temporal plane cortex (STP). Through comparative spatiotemporal analyses, we decode cell-type- and region-specific transcriptional alterations associated with chr21 abnormalities, including a disrupted inhibitory-to-excitatory balance during mid-gestational development. *RUNX1* and *APP* emerge as the most significantly dysregulated chromosome21 genes in the PFC and STP, respectively. Abnormal cortical distribution of excitatory neurons in both regions is potentially driven by dysregulated neuronal migration genes and impaired lactylation metabolism. Moreover, glial cells modulate the differentiation and migration of excitatory neurons through multiple intercellular signaling pathways. These findings provide critical insights into the pathogenesis of DS-related mid-gestational cortical abnormalities and offer valuable resources for disease modeling and development of spatiotemporally targeted therapeutic strategies.

Down syndrome (DS) is a genetic condition caused by trisomy of human chromosome 21 (Ts21)[1,2], with a global birth prevalence of approximately 1 in 779–1023 live births. This persistent prevalence underscores the complex personal, medical, and societal factors[1,3]. Individuals with DS experience a wide range of clinical features, including variable neurodevelopmental delays, cognitive impairments, and craniofacial abnormalities, along with an increased risk of congenital heart defects, autoimmune disorders, and diverse neurological conditions, such as Alzheimer's disease (AD)[4]. Among these, the most profound abnormalities observed in DS are the disruption of cortical development that fundamentally alters the structure and function of the cerebral cortex, impairing cognition and behavior[5].

The human cerebral cortex, a highly specialized and evolutionarily advanced brain structure, develops through a series of intricate

[1]Department of Anesthesiology, The First People's Hospital of Zunyi (The Third Affiliated Hospital of Zunyi Medical University), Zunyi, Guizhou, China. [2]Mental Health Center of Kunming Medical University, Kunming, Yunnan, China. [3]Institute of Neurological Disease, National-Local Joint Engineering Research Center of Translational Medicine, West China Hospital, Sichuan University, Chengdu, Sichuan, China. [4]Institute of Neuroscience, Kunming Medical University, Kunming, Yunnan, China. [5]Department of Obstetrics and Gynecology, Renmin Hospital of Wuhan University, Wuhan, Hubei, China. [6]Department of Anesthesiology, Affiliated Hospital of Zunyi Medical University, Zunyi, Guizhou, China. [7]These authors contributed equally: Rui-Ze Niu, Lu-Lu Xue, Xiao-He Tian. ✉e-mail: wangtinghua@vip.163.com; 2013103020057@whu.edu.cn; liulin.xiong@mymail.unisa.edu.au

cellular and molecular processes, including neurogenesis, neuronal migration, and cortical layering[6]. These processes underlie the functional specialization of regions such as the prefrontal cortex (PFC), which governs higher-order cognitive functions like memory, decision-making, and social behavior, and the superior temporal plane (STP), which is critical for auditory processing and language comprehension[7,8]. Dysfunction in these regions contributes significantly to the cognitive and neurodevelopmental deficits in DS. Despite progress in understanding the general mechanisms of cortical development, the specific cellular and molecular disruptions in DS, particularly during the fetal stage, remain poorly understood due to the scarcity and limited availability of DS fetal brain samples.

Cortical neurogenesis in vertebrates occurs in an inside-to-outside manner, with neuronal migration playing a pivotal role in ensuring proper positioning and cortical layer formation[9]. Aberrant neuronal migration in DS has been linked to cortical thinning, abnormal layering, and the cognitive and motor impairments characteristic of the disorder[10,11]. The molecular mechanisms underlying these developmental alterations, particularly during early fetal development, remain unclear. Recent advances in single-cell transcriptomics have provided a powerful approach to systematically decipher the cellular heterogeneity and intricate molecular events of regulating neuronal migration and development, allowing for more precise insights into the pathophysiology of DS[12]. In this study, we utilized single-nucleus RNA sequencing (snRNA-seq) to decode spatio-temporal gene expression dynamics and cellular heterogeneity in the developing cerebral cortex of fetuses with DS. By analyzing 22 mid-gestational brain samples, including the PFC and STP, from DS and control cases, we constructed a comparative cellular atlas of cortical development in DS. Our findings reveal profound alterations in neuronal migration, cortical layering, and cell-cell communication, shedding light on the molecular mechanisms underlying DS-associated cortical abnormalities during the second trimester.

## Results

### Transcriptional atlas of prefrontal and superior temporal plane cortex in fetuses with DS

To explore the cellular and molecular alterations underlying DS during fetal brain development in the second trimester, we performed snRNA-seq on 22 postmortem cortical samples (PFC and STP) from 4 fetuses with DS and 11 controls (Fig. 1a, Supplementary Data 1), with groups comparable in postmortem interval (PMI) (Supplementary Data 1). The study included 4 fetuses with DS (2 cases providing both PFC and STP samples, 1 case providing only PFC, and 1 case providing only STP), and 11 control fetuses (5 cases providing both PFC and STP samples, 4 cases providing only PFC, and 2 cases providing only STP) (Supplementary Data 1).

Although the gestational ages of our collected samples were all within the mid-gestation period, the relatively broad age range and sex imbalance between groups necessitated both global and stratified analyses to ensure data reliability (Supplementary Fig. 1). We first performed global analysis using all available samples to maximize statistical power. To account for potential confounding effects, we further conducted stratified analyses based on gestational age and sex where feasible. After QC, we retained 378,232 nuclei (156,090 from PFC; 222,142 from STP), with comparable gene and transcript counts in each cell nucleus between the two groups (Fig. 1b–d, Supplementary Fig. 2a–c). Uniform manifold approximation and projection (UMAP) indicated similar cellular architecture across brain regions (Fig. 1b, c). Based on clustering, marker gene analysis, and comparisons with previously published data, we annotated 17 high-resolution cell types (Supplementary Fig. 2d, Supplementary Data 2), including 10 excitatory neuron subtypes (ExNs, 142,134 cells, 37.6%), inhibitory neuron subtypes (InNs, 45,182, 11.9%), neuroblast (NB, 61,403, 16.2%), oligodendrocyte precursor cells (OPC, 4767, 1.3%), astrocyte subtypes

(Astro, 4743, 1.3%), microglia (Micro, 395, 0.1%), and several vascular cell types (Vas, 627, 0.2%) (Fig. 1b–d, Supplementary Data 3). Cell types were classified according to established transcriptional signatures and nomenclature consistent with recent publications[8,13,14]. Cell-type identities were well-conserved across donors. ExNs subtypes were sub-classified based on expression of cortical-specific marker genes[13,15,16] and neurodevelopmental genes (*DCX* and *STMN1*, important marker genes for immature neurons during brain development)[17–19] (Supplementary Fig. 2d). Both ImN and NB exhibit high levels of *CEP170* and tubulin family genes (Supplementary Fig. 2d, Supplementary Data 4), regulating the stability and dynamic changes of microtubules to control the migration of ImN and axonal growth[20,21]. NB cells express higher levels of *DCX* and tubulin genes, while lacking the expression of cortex-specific genes. InNs were further subclassified by ganglionic eminence origin (*LHX6* for medial ganglionic eminence (MGE); *NR2F2* for caudal ganglionic eminence (CGE))[22].

A global comparative analysis of cell populations between DS and control samples revealed significant alterations in both the PFC and STP regions, with more pronounced changes observed in the PFC (Fig. 1e–h). Stratified analyses based on gestational age further validated these findings, demonstrating at least 70% consistency with the global results (Supplementary Fig. 2e, f), thereby supporting the robustness and generalizability of our observations. Consistent with findings from the DS mouse model and studies involving the adult human PFC[14,23,24], both regions exhibited an increased InN/ExN ratio (Fig. 1f, Supplementary Data 3), with CGE-derived InNs driving this shift in PFC, and MGE-derived InNs in STP (Fig. 1g, h). Except for ExN_5/6 and ImN-L5, the cells with differential changes showed opposite variations in the two neocortices (Fig. 1g, h). In addition, case-control analysis (cacoa)[25] confirmed regional differences in cellular changes in the DS brain (Supplementary Fig. 3a). Moreover, we observed an increased transcriptional heterogeneity in the PFC of fetuses with DS, which exhibited significantly higher inter-individual expression variability compared to the STP region (Supplementary Fig. 3b, c), suggesting early region-specific dysregulation. This variability may underlie phenotypic heterogeneity in DS, as the PFC governs higher-order functions such as cognition and behavior[26].

Aggregate analysis of 621 chromosome 21 (chr21) genes showed significant expression elevation in fetuses with DS (mean fold change = 1.48 for STP and 0.27 for PFC, $P$adj < 0.01; Fig. 1i), with stratified analyses showing great consistency of variation trends with the global findings (Supplementary Fig. 3d). While our bulk analysis confirmed significant upregulation of chr21 gene sets (Fig. 1i), we observed the established pattern of partial dosage compensation: 12.1% (STP) and 0% (PFC) of genes showed ≥2-fold increase, 39.03% and 0.45% exhibited >1-fold change, and 54.77% and 19.82% were variably expressed – aligning with prior reports of chr21 regulation with variable dosage sensitivity in DS[1,27].

### Systematic differential analysis of gene expression

We identified differentially expressed genes (DEGs) between DS and control samples across the major cell types (Fig. 2a, b, Supplementary Data 5). Notably, STP cells exhibited widespread upregulation (>90% of DEGs), contrasting with the PFC. Among these cell types, ExN L5/6 and ExN L3-5 populations demonstrated the highest susceptibility to disease-related transcriptomic alterations, followed by NB (Fig. 2b, c). Only a minor fraction of DEGs overlapped between PFC and STP (Fig. 2c). To explore sex-related effects, we stratified the DEGs by sex. Given the limited number of male samples, subgroup analyses were restricted to female fetuses (2 DS females vs. 4 controls). These female-only results showed strong concordance (>73% in PFC cell types and >62% in most STP cell types, except for Astro) with the full dataset (Supplementary Data 6). In the PFC, 60% of cell types showed at least 60% DEG overlap between males and females (Supplementary Fig. 4a), while the STP exhibited more pronounced sex-related differences,

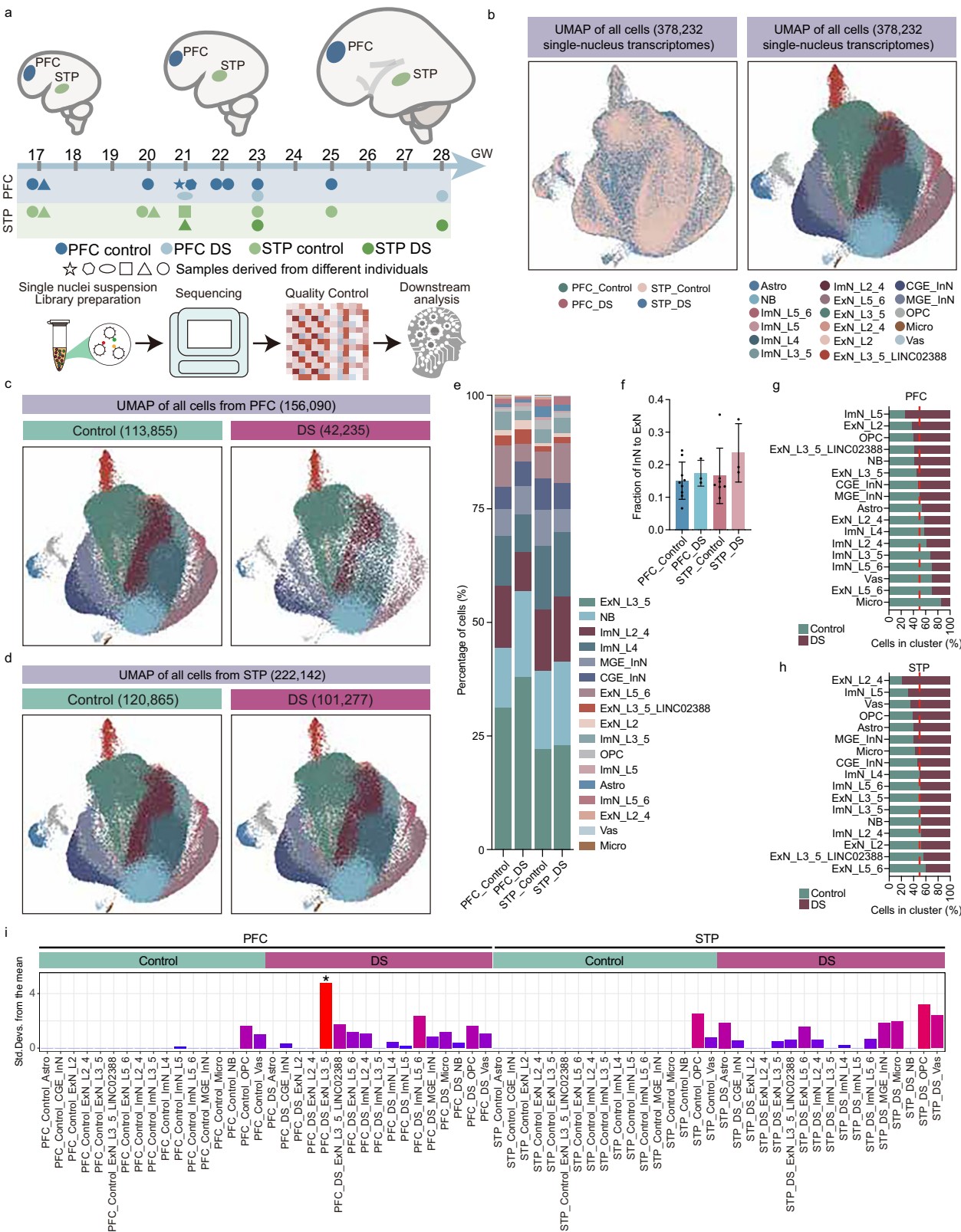

with DEG consistency between sexes dropping to as low as 49.7% (Supplementary Fig. 4b). When compared with bulk RNA-seq data from DS PFC tissues[28], 69.4% of genes showed consistent expression change trends (Supplementary Fig. 4c). Further analyses across stratified datasets revealed that the extent to which cell types are influenced by gestational age varied (Supplementary Data 7). Nevertheless, disease-sensitive populations such as ExN L5/6, ExN L3-5, and NB

exhibited high consistency across global and stratified analyses−≥82% in PFC (Supplementary Data 7). In contrast, non-neuronal populations showed a much smaller number of DEGs, potentially due to lower statistical power associated with their low abundance. These comparative observations on the number and dominant directionality of DEGs highlight the heterogeneous responses of different brain regions and cell types to DS pathology. Further analysis revealed that the STP

**Fig. 1 | Transcriptional atlas of prefrontal and supratemporal plane cortex in fetuses with DS. a** Workflow of sample collection, data generation, integration, and annotation for the fetal brains' snRNA-seq datasets. **b** UMAP of cells grouped by disease conditions (left) and cell types (right), showing clustering across PFC and STP. **c** UMAP visualization of cell type from control and DS in PFC. **d** UMAP visualization of cell type from control and DS in STP. **e** The cellular composition of PFC and STP illustrates a comparative analysis of cell proportions between DS and control groups. **f** InN/ExN ratios in control and DS brains from PFC and STP. $n = 9$ biological replicates for PFC_Control, $n = 3$ biological replicates for PFC_DS, $n = 7$ biological replicates for STP_Control, $n = 3$ biological replicates for STP_DS. Two-sided paired Student's $t$-test. Data are shown as mean ± SEM. Source data are provided as a Source Data file. Bar plot showing the comparison of various cell types in both PFC (**g**) and STP (**h**) across the control and DS. **i** Enrichment levels of chr21 genes in PFC and STP for both control and DS groups. Asterisks denote the Benjamini–Hochberg (BH)-corrected $p$-value < 0.05 calculated using EWCE (Permutation Test), $p = 0.0068$. Source data are provided as a Source Data file. PFC prefrontal cortex, STP supratemporal plane, ExNs excitatory neurons, NB neuroblast, InNs inhibitory neurons, MGE/CGE medial or caudal ganglionic eminence, Astro astrocytes, Micro microglia, OPC oligodendrocyte progenitor cells, Vas vasculature-associated cells.

transcriptome was more susceptible to chromosomal abnormalities than the PFC, with a larger number of DEGs detected in at least 10 STP cell types (192 genes) compared to the PFC (30 genes) (Supplementary Data 5). Key DEGs included *RUNX1*, *ANKRD36B*, and *ANKRD36C* in the PFC, and *APP*, *CD24*, and *STMN2* in the STP (Supplementary Data 5). These findings suggest that different brain regions exhibit distinct molecular vulnerabilities to DS pathology.

We examined the chromosomal distribution of DEGs and found that most were localized on chr1-8, with a smaller subset on chr21 (Supplementary Fig. 5a, b). Unlike in adult individuals with DS[14], chr21 DEGs were predominantly upregulated in neurons of fetuses with DS, rather than in microglia or endothelial cells (Fig. 2d). Notably, stratified analyses based on gestational age revealed a high degree of concordance with the global dataset—at least 80% of chr21 DEGs were consistently identified across datasets (Supplementary Data 8). This suggests robust differential expression of chr21 genes in DS, minimally influenced by gestational age variability. Nine shared chr21 DEGs, including *DSCAM*, *APP*, *CXADR*, *SON*, *KCNJ6*, *TIAM1*, *GRIK1*, *MIR99AHG*, and *NCAM2*, showed significant changes in both the PFC and STP (Fig. 2d). *RUNX1* and *APP* were the most significantly altered genes in the PFC and STP, respectively. Virtual knockout analysis implicated *RUNX1* in regulating multiple tubulin family genes and *STMN1/2*, which are involved in neuronal development and axon regeneration, while *APP*-targeted genes were primarily linked to dendritic transport and neuroepithelial cell differentiation (Supplementary Fig. 5c, d, Supplementary Data 9).

To contextualize fetal findings in later development, we compared PFC data from young and aged individuals with DS[14] (Supplementary Data 10), where 8 chr21 DEGs in ExNs and 5 in InNs persist across development stages. DEGs overlap declined with age, with ExN DEGs showing early disruption, while InN changes increased later (Fig. 2e).

A previous study indicated that brain organoids derived from patients with DS can partially model abnormal neurogenesis in DS[29]. We compared fetal signatures with in vitro cortical organoid differentiation trajectories and found that the organoids shared more overlapping DEGs with the STP, supporting their use as a model for DS STP research (Fig. 2f). Organoids cultured for 30 days showed more similarity to the brain than those cultured for 70 days. Shared genes *TIAM1* and *APP* showed significant changes in both the brain and organoids, suggesting their critical role in DS brain function. Further integration of snRNA-eq data from organoids indicated that DS-derived organoids more accurately simulate the neuronal differentiation status of the STP, such as the consistent cell count of InN1-3 in both fetal brain and organoids (Fig. 2g, h). However, current organoid models only simulate a small fraction of the abnormalities found in the PFC or STP of individuals with DS.

Beyond the brain, DS pathology extends to peripheral tissues. Transcriptomic analysis of peripheral blood from a large clinical cohort[30], comprising 304 individuals with DS (163 males, 141 females) and 96 euploid controls (44 males, 52 females), revealed significant alterations in the transcriptome of peripheral blood in individuals with DS, accompanied by upregulation of most genes encoded on chr21. Notably, peripheral blood exhibited a significant upregulation of chr21 genes with a greater number of DEGs compared to the neocortices (Fig. 2i, Supplementary Data 11), suggesting that peripheral blood is more susceptible to chr21 gene triplication than brain tissue. Nevertheless, there was minimal overlap between blood and brain DEGs (Fig. 2i), reflecting the tissue-specific nature of DS pathology. One notable exception was *SON*, a chr21 DEGs significantly changed in both the neocortices and peripheral blood (Fig. 2i). Research indicates that the de novo mutations of *SON* can affect the key genes (*TUBG1* and *HDAC6*) crucial for neuronal migration and cortical organization, disrupt the RNA splicing of genes essential for brain development and metabolism, and lead to intellectual disability syndrome[31]. These findings suggest that peripheral blood may serve as a useful proxy for studying certain molecular aspects of DS and identifying potential biomarkers.

## Disrupted development of ExN subtypes in the neocortices of fetuses with DS

Perturbations in neuronal migration and differentiation are thought to contribute to cortical abnormalities in DS, but their effects in the human fetal brain remain poorly defined. Previous studies in mice have shown that gain-of-function (GOF) mutations in *DSCAM* and *DSCAML1* impair forebrain cortex development and neuronal migration during fetal stages[32]. To investigate these processes in humans, we performed subclustering of ExNs in the PFC, identifying layer-specific subtypes (Fig. 3a, Supplementary Fig. 2d), marked by *CUX2* (L2), *RORB* (L4), and *PCP4* (L5) expression[15,16]. Analysis of cell-type proportions across late second trimester revealed broadly similar developmental trajectories between control and DS groups in both PFC and STP, but several subpopulations showed altered abundance, particularly around GW23 (Fig. 3b). Immature ExNs expressing *DCX* and tubulin, as well as mature *CUX2*+ L2 and L2–4 neurons, were differentially represented in DS (Fig. 3b). *CUX2*-enriched L2 neurons have been implicated in neurodevelopmental and neuropsychiatric disorders such as AD, Multiple Sclerosis (MS), and Schizophrenia (SCZ)[25,33].

To validate these results, we examined the cortical distribution of upper-layer (*SATB2*+*FOXP2*−) and deep-layer (*SATB2*+*FOXP2*+) neurons at GW23 using immunofluorescence (Fig. 3c, d). In controls, *FOXP2*+ cells predominantly localized to the deep cortical layers, and *SATB2*+ cells to middle-upper cortical layers (Fig. 3c), consistent with known inside-out cortical layering[8]. In DS fetal brains, a reduction in upper-layer neurons and increased deep-layer neurons abundance were observed in PFC and STP regions (Fig. 3c, d). Bin-based counting analysis showed that *FOXP2*+ cells and *SATB2*+ cells exhibited differential distributions between control and DS in various cortical layers, which aligns with the results from snRNA-seq (Fig. 3e, f). These findings support the impaired inside-out migration pattern of neocortical neurons in DS, a hallmark of normal cortical development.

Further analysis identified 52 DEGs in PFC and 65 DEGs in STP related to cell migration (Fig. 4a, Supplementary Data 12). To evaluate the robustness of these findings, we performed Jaccard similarity analyses of overlapped migration-related DEGs between each stratified subgroup and the full GW17−28 dataset, revealing consistency rates of at least 73% in PFC and 66% in STP (Supplementary Fig. 6a). The protein-protein interaction (PPI) network indicated enrichment in

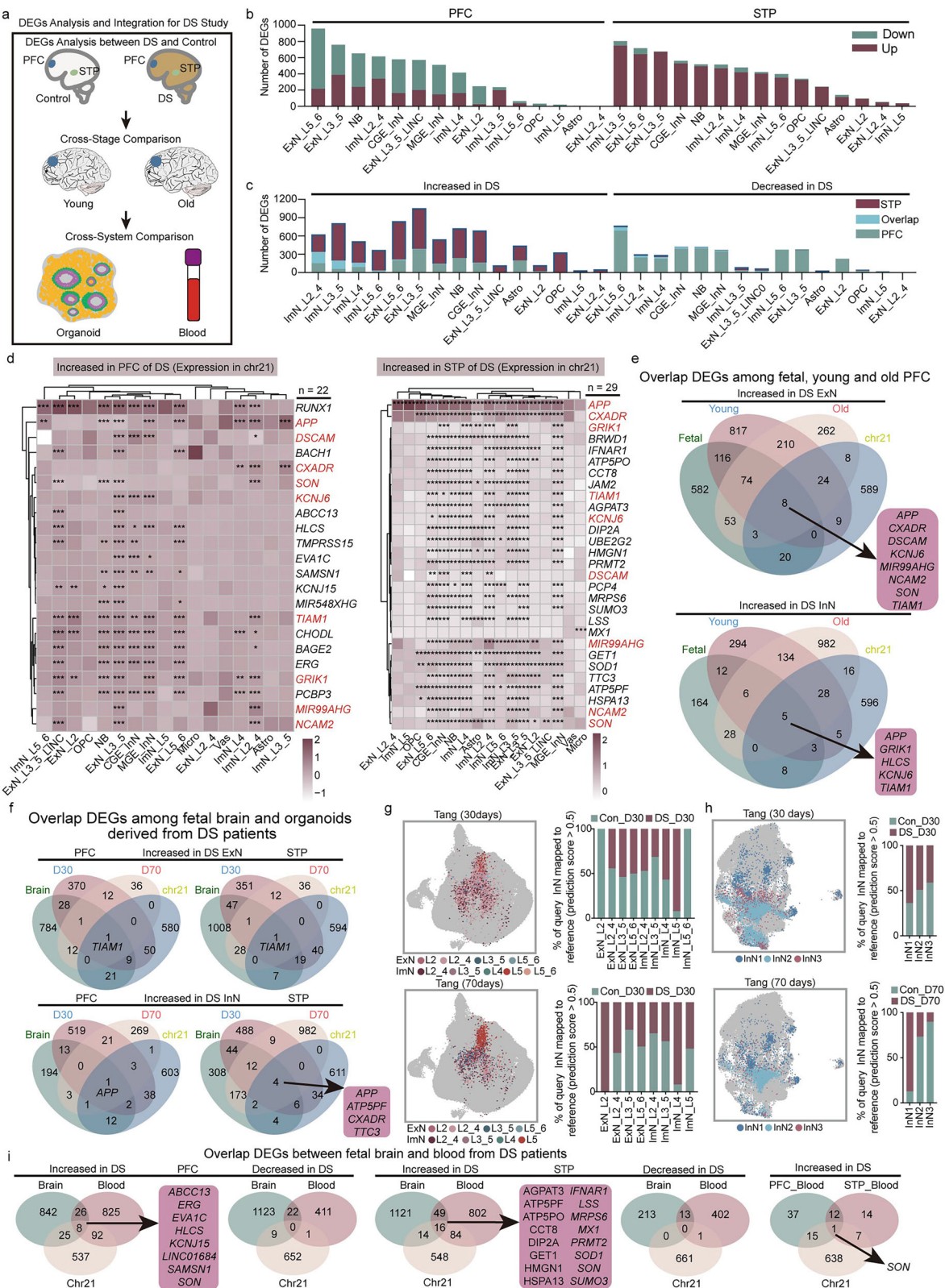

pathways involved in neuronal migration, cell dynamics, cell positioning, and axon development (Supplementary Fig. 6b–e). Cytotrace analysis and Monocle analysis revealed that DS neurons were transcriptionally less mature and exhibited delayed differentiation and migration (Fig. 4b, Supplementary Fig. 7a, b). In controls, migration-related genes displayed distinct temporal expression patterns in the late second trimester, whereas DS brains exhibited enrichment of

neuronal migration genes around GW23, suggesting dysregulated timing (Supplementary Fig. 7c, d). This phenomenon was observed in both the PFC and STP. Additionally, genes associated with immune cell migration were highly expressed in control brains at GW23, suggesting synchronized but distinct migration processes between neurons and immune cells during cortical development. This synchronization appeared disrupted in DS, where dysregulated migration genes may

**Fig. 2 | Transcriptional differences of distinct cell types in the PFC and STP of fetuses with DS. a** Number of DEGs of PFC and STP across different cell types. **b** Number of DEGs of PFC and STP across different cell types. Differential expression was assessed using a two-sided likelihood ratio test based on MAST, with covariate adjustment for library size, mitochondrial gene content, and sex. *P*-values were corrected for multiple comparisons using the BH method. **c** Number of overlapping DEGs of PFC and STP across different cell types. **d** Expression heatmap of chr21-related DEGs in PFC and STP across different cell types. DEGs were assessed using a two-sided likelihood ratio test based on MAST. *P*-values were corrected for multiple comparisons using the BH method. *FDR < 0.05, **FDR < 0.01, ***FDR < 0.001. **e** Venn diagram showing the number of DEGs in ExNs and InNs of PFC at fetal, young, and old stages. **f** Venn diagram showing the number of DEGs in ExNs and InNs of PFC between fetal brain and organoids from patients with DS[29]. UMAP projection of ExNs (**g**) and InNs (**h**) in two query datasets of DS-derived brain organoid cultures to our in vivo ExNs and InNs reference map. Colors represent the assigned subpopulations, and intensity represents the prediction score for each query cell. Bar plots show the proportions of query cells mapped to our in vivo ExNs and InNs reference. **i** Venn diagram showing the number of DEGs between PFC and peripheral blood[30]. Source data are provided as a Source Data file.

interfere with the interaction between innate immune cells and brain parenchymal cells, potentially contributing to cortical development abnormalities.

To evaluate the diagnostic potential of migration-related DEGs, we constructed machine learning models using these genes as feature variables. We found that these genes were able to accurately distinguish the neuron states across different conditions, achieving 78% and 71% accuracy in the PFC and STP, respectively (Fig. 4c–e). Besides, our predictive model maintained about 70% accuracy when applied to stratified data (Supplementary Fig. 8a). Furthermore, by integrating the data from two normal PFC datasets (GSE168408 includes one sample at GW22 and one at GW24; GSE204684 includes one sample at GW22 and one at GW24), we found that the prediction accuracy of our model for normal cells could reach 99% (Supplementary Fig. 8a). In these models, *TUBA1A*, *TUBB2B*, and *TUBB2A* were the highly contributing genes in both PFC and STP (Fig. 4c–e). These genes exhibited significant expression differences during cortical development (Fig. 4f), and immunostaining confirmed abnormal distribution of *TUBB2B*+*NeuN*+*SATB2*+ neurons in DS cortex at GW23 (Fig. 4g–i, Supplementary Fig. 8b). Finally, virtual knockout analysis of DS-associated DEGs identified dysregulated chr21 genes in both PFC and STP, which could regulate tubulin-mediated neuronal microtubule dynamics and axon formation (Fig. 4j, Supplementary Data 13, Supplementary Data 14). These findings emphasize the critical role of microtubule-associated dysregulation in disrupted cortical development in the late second trimester of DS.

## DS neocortical cells exhibit significant metabolic alterations

Studies have shown that abnormal levels of glycolytic and tricarboxylic acid (TCA) cycle metabolites, as well as altered insulin signaling, in the brain tissue of patients with DS suggest their potential role in cognitive deficits[34]. Through single-cell transcriptomic metabolic analysis, we observed widespread and significant metabolic changes across cell types within the PFC and STP of DS (Fig. 5a). Notably, disruptions in neurotransmitter-related metabolites, such as GABA, glutamate, and glutamine, form the basis for impaired signaling and synaptic transmission between neurons (Fig. 5a). These metabolic abnormalities were further associated with dysfunctions in neuronal development, migration, neurotransmitter synthesis, and synaptic plasticity. Research has indicated that ribosome stalling—harmful to the organism—tends to favor the accumulation of proline, arginine, and lysine[35]. Consistently, we found significant upregulation of proline and arginine in all cell types in the PFC NB and STP (Fig. 5a). These findings suggest that, beyond transcriptomic changes, alterations in ribosome-associated metabolites further exacerbate cortical developmental abnormalities in DS. Such metabolic disruptions likely interfere with the translation of key functional proteins or enzymes, thereby contributing to the pathogenesis of DS.

Recent research has highlighted the role of protein lactylation, a post-translational modification of lysine residues in histones and non-histone proteins, as a critical regulator of neuronal function[36,37]. Increased hypoxia and glycolysis have been shown to enhance protein lactylation levels in various cell types[38]. It has been reported that *HDAC6* catalyzes the lactylation of α-tubulin in a concentration-

dependent manner, promoting the dynamics of neuronal microtubules[39]. In our analysis, both global and hierarchical analyses demonstrated widespread changes in lactate levels in cells of the PFC and STP (Fig. 5b, Supplementary Fig. 9a–c, Supplementary Fig. 10a, b), providing a basis for abnormal protein lactylation modifications in neurons. Mass spectrometry analysis identified 285 peptides in cortical neurons subject to lactylation modifications[39]. Among these, we observed significant changes in 48 and 44 lactylation-regulated molecules in the PFC and STP, respectively, including down-regulation of α-tubulin (Fig. 5c, Supplementary Data 15). The PPI network showed strong interconnections between these genes, which are primarily associated with neuronal migration and microtubule dynamics (Fig. 5d). Jaccard similarity analyses compared overlapped lactylation-regulated molecules among stratified subgroups to the full GW17–28 dataset, revealing molecular consistency rates of at least 67% in PFC and 58% in STP (Supplementary Fig. 10c). These findings establish a potential link between DS-associated cortical abnormalities and extensive lactylation dysregulation, particularly involving α-tubulin (Fig. 5e).

## Dysplasia of InNs in DS affects ExN migration through cellular communication

Abnormal proportions and subtypes of InNs and ExNs in the PFC of adult patients with DS have been well-documented[14]. Through sub-population analysis of InNs in the PFC and STP, we identified 10 distinct subtypes (Fig. 6a). InN1, InN5, and InN8 express *NR2F1* and *NR2F2*, which are derived from the caudal ganglionic eminence (CGE). InN3, InN4, InN6, InN9, and InN10 express *LHX6* and *SOX6*, originating from the MGE, while InN2 and InN7 express *MEIS2*, *FOXP1*, and *FOXP2*, derived from the lateral ganglionic eminence (LGE) (Fig. 6b). Consistent with previous studies[15,40], the cell populations enriched with *CCK*, *CALB2*, and *VIP* are derived from the CGE, while *SATB1* and *SST*-enriched cells originate from the MGE. Studies have shown that Reelin (*RELN*) plays a crucial role in neuronal migration and cortical lamination, and its reduced expression may impair neuronal migration in the hippocampus, leading to neural circuit abnormalities that impact cognitive functions such as memory, learning, and language comprehension[41]. Our normalized cell counts showed a reduction in the *RELN*+ (InN6 and InN9) cell populations (Fig. 6c), suggesting a potential link to DS-related cognitive dysfunction (Fig. 6d). Cytotrace analysis revealed that all InNs are differentiated cells, with only subtle differentiation differences (Fig. 6e). Monocle pseudotime analysis indicated significant differentiation differences between InN subtypes across fetal ages, cell types, brain regions, and experimental groups (Fig. 6f, g). Notably, transcription factors *ARX*, *CUX1*, and *SOX11* exhibited significant differential expression between control and DS InNs, highlighting potential regulatory dysfunctions (Fig. 6h).

To explore how InNs regulate ExNs, we employed CellChat, NicheNet, and NeuroChat to analyze cellular communication. CellChat and NeuroChat analyses revealed opposing changes in cellular communication within the PFC and STP in DS compared to the controls (Supplementary Fig. 11a, b). CellChat analysis indicated that *NRG, PTN*, and *SEMA3* were the most abundant and intense forms of cellular communication (Supplementary Fig. 11c). NeuroChat, a neuron-

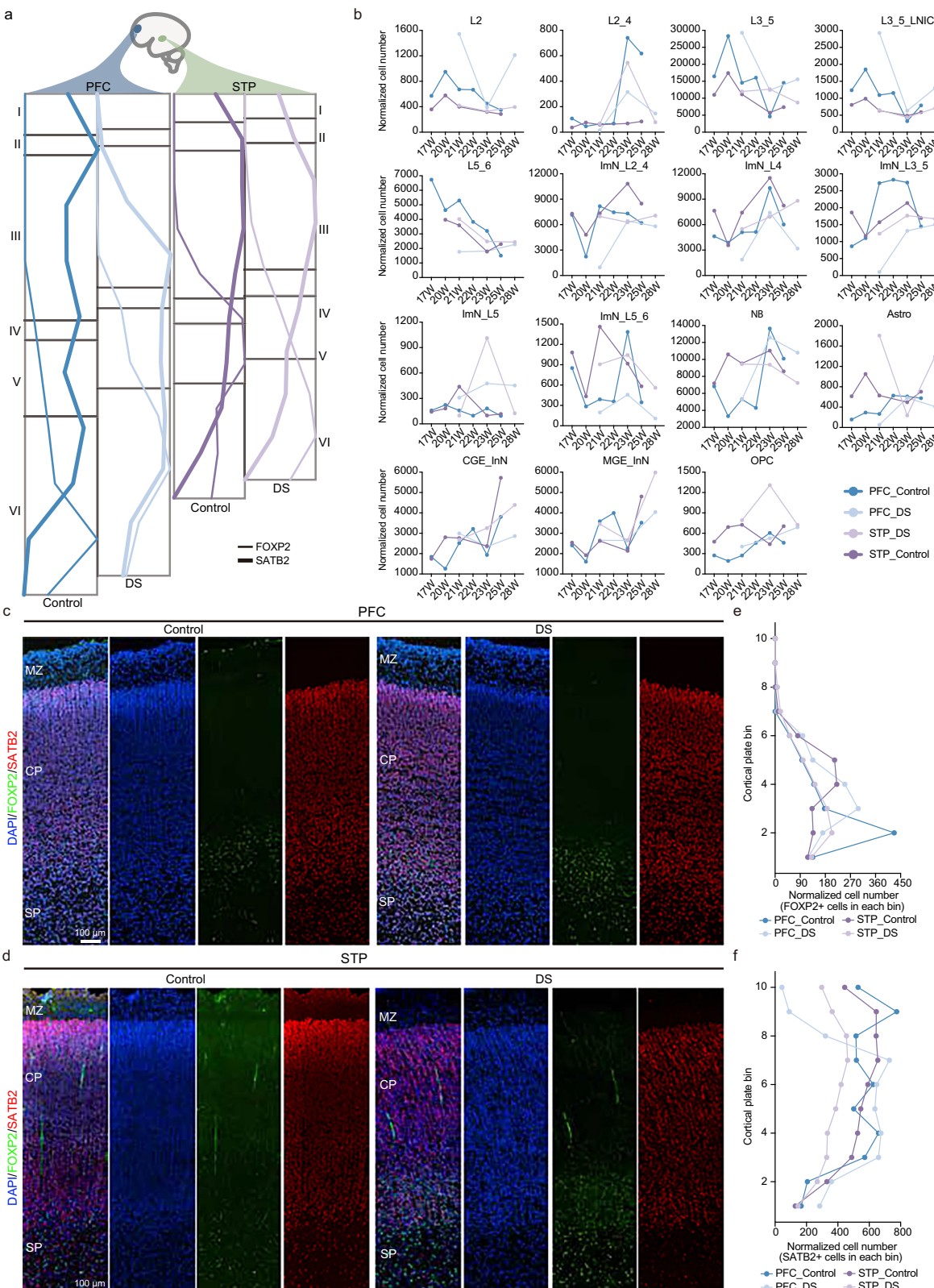

**Fig. 3 | The changes of ExN subgroups in PFC and STP with gestational age.** **a** The model diagram indicates that the PFC and STP in individuals with DS exhibit both overall and localized differential changes. **b** The trends of standardized cell numbers of different cell types from PFC and STP in the control and DS groups with respect to gestational age. **c**, **d** Representative microscopic fields of *FOXP2* (green)/ *SATB2* (red) positive cells in the PFC (**b**) and STP (**c**) from GW23 of control and DS. Blue, DAPI. Scale bar, 100 μm. **e**, **f** Quantification of *FOXP2* (green)/*SATB2* (red) positive cells in PFC and STP from Control and DS group. *n* = 3–4 sections from 3 DS samples or 3 control samples. CP cortical plate, SP subplate zone, MZ marginal zone. Source data are provided as a Source Data file.

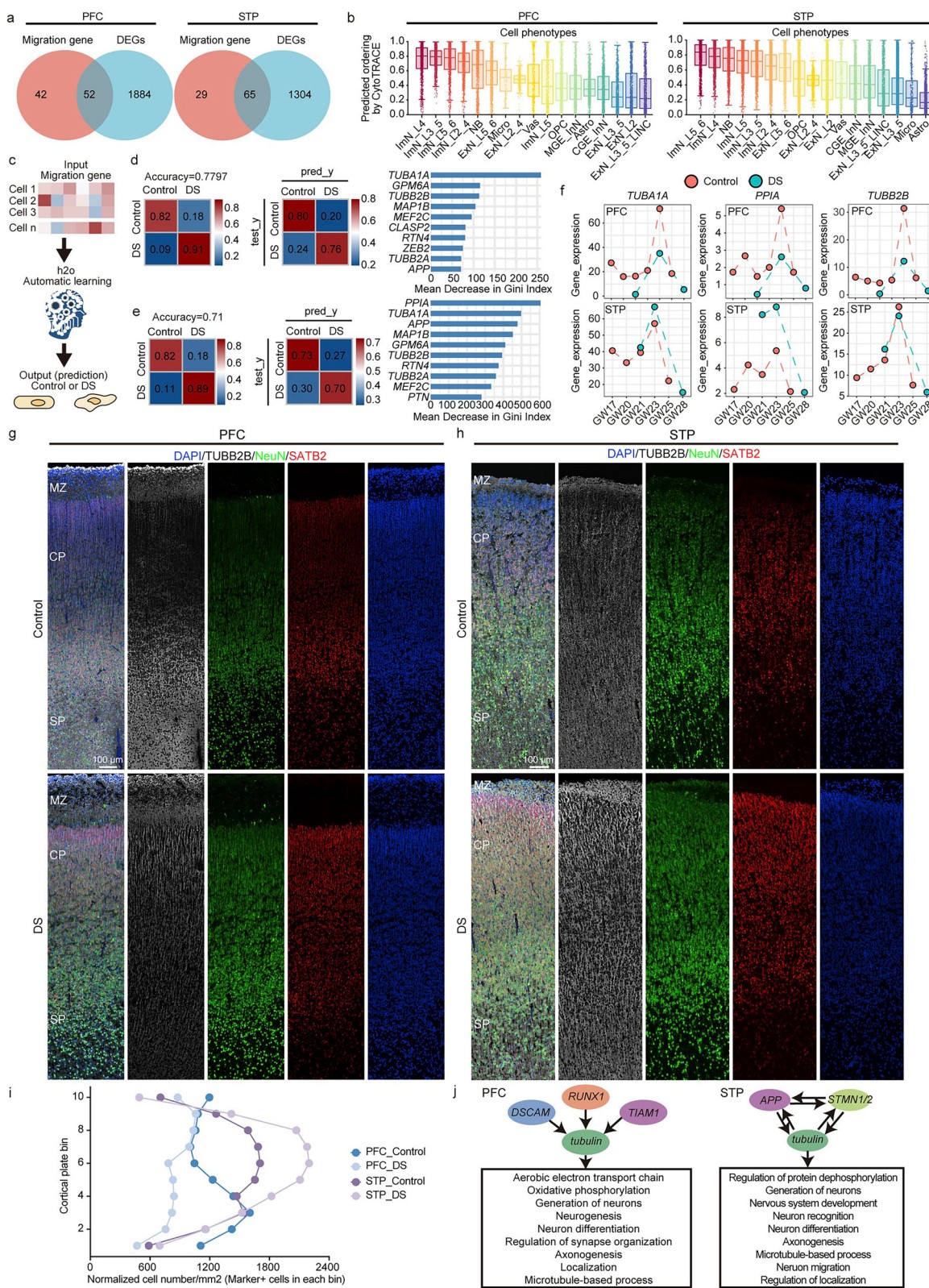

specific cellular communication analysis, identified *NLGN1*-related signaling as the most intense and significantly altered form of communication (Supplementary Fig. 11d). Further investigation showed that cellular communication between InNs and ExNs was reduced in the PFC but increased in the STP (Fig. 6i). Specifically, we observed significant reduction in *NRXN1-NLGN1*, *NRXN2-NLGN1*, and *NRXN3-NLGN1* communication in the PFC, whereas these interactions were

significantly elevated in the STP (Supplementary Fig. 11e). This discrepancy underscores the region-specific impacts of DS pathology on cortical development. Ligand-receptor analysis further identified *NLGN2* and *PTN* from InNs as key regulators of ExN development and migration in DS. These ligands target multiple ExN genes, including *RUNX1*, *STMN1*, *TUBB*, and *TUBB2B*, which are critical for microtubule dynamics and neuronal migration (Fig. 6j). Together, these findings

**Fig. 4 | Abnormal expression of neuronal migration genes in the PFC and STP of patients with DS. a** The Venn diagram shows abnormal expression of multiple neuron migration genes. **b** Box plots show the predicted ordering of cell phenotypes by CytoTRACE in the PFC and STP. For the PFC (14,624 cells) and STP (18,154 cells), 1000 and 1200 cells were randomly selected from each cell type, respectively. Box plots indicate the median, interquartile range (25th–75th percentiles), and whiskers extend to 1.5 times the interquartile range (IQR). **c** A schematic diagram of machine learning for predicting cell state by cell migration genes. **d, e** Predictive models and feature importance of migration gene for cell state. Left, Confusion matrices of disease prediction performance for the training set. Model accuracy values are noted for each model. Middle, Confusion matrices of disease prediction performance for the test set. Right, Feature importance analysis showing the mean decrease in Gini index for each gene, indicating their relative importance in predicting cell states. **f** The expression changes of *TUBB1A*, *PPIA*, and *TUBB2B* with gestational age in the control and DS groups of PFC and STP. **g, h** Representative microscopic fields of *TUBB2B* (White)/ *NeuN* (green) /*SATB2* (red) positive cells in the PFC (**g**) and STP (**h**) from GW23 control and DS. Blue, DAPI. Scale bar, 100 μm. **i** Quantification of *TUBB2B* (White)/*NeuN* (green) /*SATB2* (red) triple-positive cells in PFC and STP from Control and DS group. *n* = 3 per group. **j** The chr21 DEGs regulated the development and migration of neurons in the PFC and STP through tubulin. CP cortical plate, SP subplate zone, MZ marginal zone. Source data are provided as a Source Data file.

suggest that intrinsic differentiation abnormalities in InNs, combined with altered cellular communication pathways, disrupt ExN migration and cortical development in DS (Fig. 6k).

## Glial dysfunction in DS affects neuronal development and migration

The differentiation of OPCs into oligodendrocytes is critical for neuronal developmental processes, such as migration, proliferation, and differentiation[42]. In this study, subpopulation analysis identified four distinct OPC subtypes in the PFC and STP of fetuses with DS (Fig. 7a). OPC3, expressing *OLIG1* and *OLIG2*, represents the major source of mature oligodendrocytes in adults (Fig. 7b). OPC4, which expresses *BCAS1*, *MPB*, and *PLP1*, corresponds to immature oligodendrocytes (Fig. 7b). Notably, the numbers of OPC1 and OPC2 were found to be oppositely regulated in the PFC and STP of DS, while both OPC3 and OPC4 populations were significantly reduced in the DS neocortices (Fig. 7c). Immunofluorescence staining confirmed these findings, with a notable decrease of *BCAS1*+*OLIG2*+ double-positive cells in the DS neocortex (Fig. 7d, e).

Analysis of Astro in the PFC and STP identified four distinct subtypes (Fig. 7f). Astro1 and Astro2, which express *GFAP, AQP4, ID2*, and *GPC5, DPP10*, represent the main Astro populations (Fig. 7g). Astro3, expressing *SOX2* and *DCX*, potentially consisting of neural progenitor cells[43] (Fig. 7g). Astro4, expressing *ASCL1, CDK1, TOP2A*, and *HMGB2*, consists of potential intermediate neural progenitors[44,45] (Fig. 7g). In DS, Astro1 and Astro3 populations were increased in both the PFC and STP, while Astro2 and Astro4 populations are decreased (Fig. 7h). The molecules *GPC5* and *DPP10*, which are expressed in Astro, are known to promote neuronal growth and synapse formation[46,47]. Immunofluorescence staining also validated these snRNA-seq results, showing a significant increase in *ID2*+*GFAP*+ double-positive cells in the DS neocortices (Fig. 7i, j). These shifts in cell population and dynamics and gene expression among neural and intermediate progenitor cells suggest abnormalities in the differentiation of these cells in fetuses with DS.

To investigate how glial dysfunction affects ExN development, we utilized NicheNet analysis to identify ligand-receptor interactions between OPCs, Astro, and ExNs. Consistent with the results from CellChat and NeuroChat, both OPCs and Astro were found to regulate the expression of downstream target genes in ExNs by targeting receptors such as *NLGN1* and *ERBB4* (Fig. 7k, l). Further analysis of the downstream target genes of *NLGN1* and *ERBB4* revealed that OPCs and Astro in the PFC affect the expression of *RUNX1* in ExNs, which subsequently influences tubulin expression (Fig. 7m, Supplementary Data 16). In contrast, in the STP, OPCs and Astro predominantly affect the expression of *DSCAM* in ExNs, a gene involved in neuronal differentiation and recognition (Fig. 7m). These findings emphasize the region-specific roles of glial cells in modulating ExN development in DS.

## The abnormality of the DS transcriptome is associated with neural function and related disease susceptibility

Using Expression-Weighted Cell-type Enrichment (EWCE) analysis, we assessed the enrichment of 24 neuropsychiatric disease risk genes in the DS snRNA-seq data (Supplementary Data 17). Compared with PFC, STP exhibited a higher association with disease risk genes (Fig. 8a, Supplementary Fig. 12a). Neuropsychiatric disorders demonstrate regional specificity. Risk genes of alcohol dependence (ALD) and multiple sclerosis (MS) are mainly enriched in the PFC, while those of Tourette syndrome (TS), AD, and Parkinson's disease (PD) are predominantly enriched in the STP. Bipolar disorder (BIP) and schizophrenia (SCZ) showed significant enrichment in both the PFC and STP, implying the complexity of the mechanisms and the heterogeneity of these diseases (Supplementary Fig. 12a). When analyzing disease by cell type, we found that STP cells are enriched with a greater number of disease-associated genes (Supplementary Fig. 12b). Stratified analyses based on gestational age yielded a high level of concordance (≥75%) in disease susceptibility compared with the global dataset (Supplementary Figs. 13–15). These outcomes underscore heightened disease susceptibility of the STP region in DS brains. Prior work has documented this region susceptibility pattern in neuropathological studies, showing a progressive volume reduction of STP in patients with schizophrenia[48–51]. In the DS transcriptome, genes associated with mental disorders such as schizophrenia, depression, and autism were abnormally expressed, potentially predisposing individuals to these conditions in later life[52]. As anticipated, genes related to aging and memory disorders were significantly enriched in the cells of the DS group in STP (Supplementary Figs. 13–15), supporting the symptoms of premature aging in DS[53–55]. Further application of single-cell Disease Risk Score (scDRS) analysis evaluated the degree of association between common mental disorders and neurofunctions in different cell types of the neocortices. We observed significant reductions in intelligence (INT), educational attainment (ECOL and EY), and verbal numeric reasoning (VNR) abilities in cells of the DS group (Fig. 8b). Conversely, cells associated with AD and MS were significantly increased in the DS group (Fig. 8b).

To further explore the connection between differential cell migration genes in ExNs and neuropsychiatric disorders, we conducted virtual knockout analysis to identify the regulatory genes associated with each migration-related gene. The resulting gene module, comprising cell migration genes and their corresponding regulatory genes, was designated as the DS-related cell migration gene module (DSMM). By examining the associations between these DSMM and various diseases, we found that STP exhibited a correlation with a greater number of disorders (Fig. 8c, Supplementary Data 18). In the PFC, the most prominent associations with aging-related genes were found, followed by those related to anxiety, ALD, SCZ, and major depressive disorder (MDD) (Fig. 8c). In the STP, the majority of associated genes were linked to aging and narcolepsy, followed by anxiety, SCZ, and MDD (Fig. 8d). These results emphasize the differential roles of PFC and STP cells in disease susceptibility, providing a framework for understanding the distinct contributions of different brain regions and cell types to DS pathology.

## Discussion

In this study, by leveraging snRNA-seq data from rare and valuable mid-gestational brain tissues, we observed an imbalance between InNs

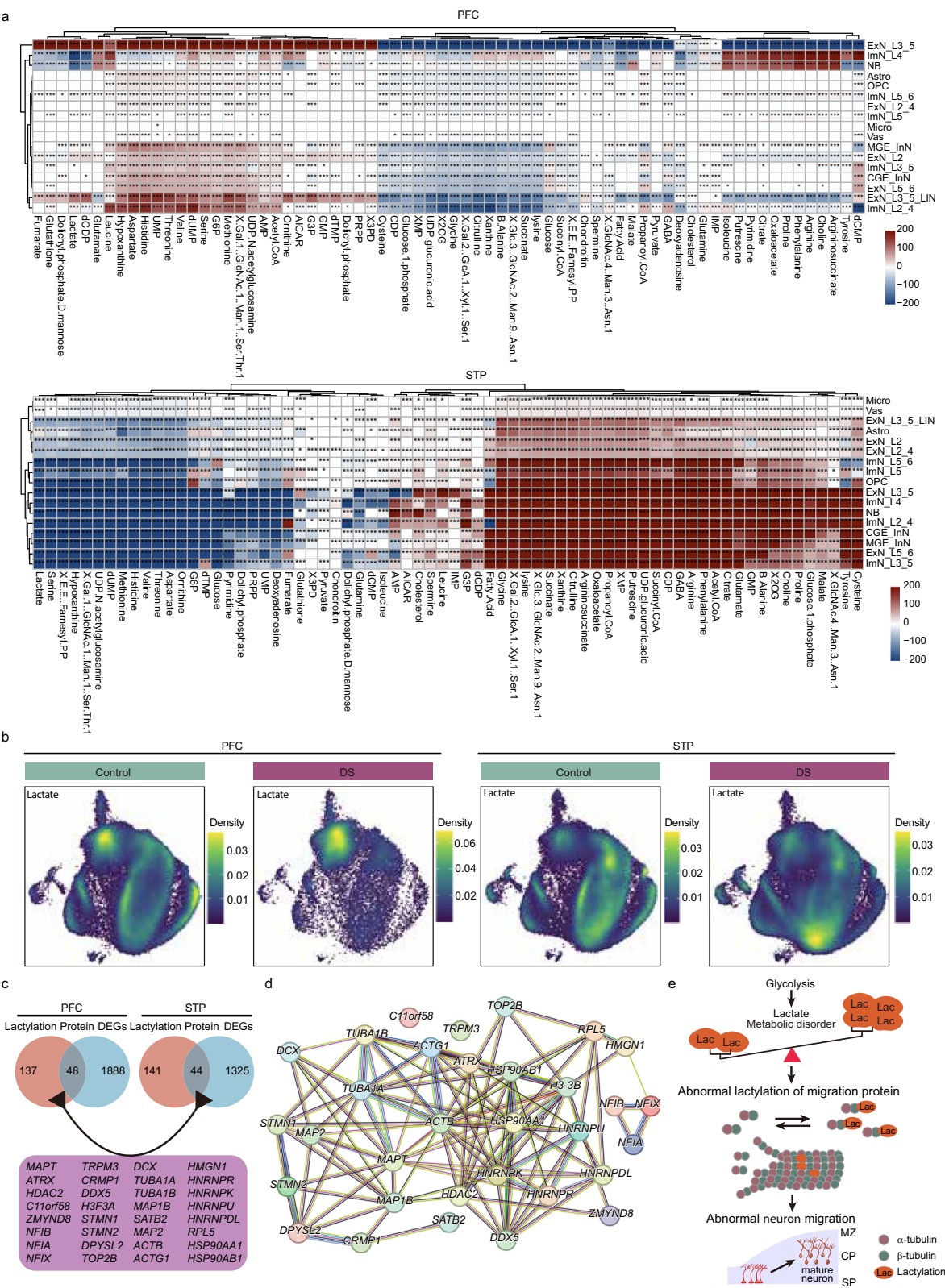

**Fig. 5 | Extensive metabolic abnormalities occur in the PFC and STP of patients with DS. a** Sixty-eight metabolites showed differential expression in different cell types of PFC and STP (two-sided Wilcoxon rank-sum test, *$p < 0.05$, **$p < 0.01$, ***$p < 0.001$). Two-sided unpaired Wilcoxon rank-sum test. *P*-values were adjusted for multiple comparisons using the BH method. Red indicates upregulation in DS, and blue indicates downregulation. **b** UMAP shows the expression density of lactate-related genes in different groups of different brain regions. **c** Venn diagram showing DEGs (FDR-adjusted *p*-value < 0.05, |log2FC| > 0.25) associated with lactylated proteins in the PFC and STP. DEGs were assessed using a two-sided

likelihood ratio test based on MAST. *P*-values were corrected for multiple comparisons using the BH method. **d** PPI network of 32 differentially changed lactylated molecules. **e** A proposed model for the abnormal regulation of migration protein lactylation in DS fetal brain. Glycolysis is crucial for energy generation in cells and results in the production of lactate. Multiple differentially expressed cell migration proteins in DS neurons are regulated by lactylation in a concentration-dependent manner and are involved in the development and migration of cortical neurons. Source data are provided as a Source Data file.

and ExNs in DS, characterized by an increased ratio of InNs during early brain development that persists into adulthood. The disruption of neuronal homeostasis is associated with significant alterations in genes critical for neuronal migration and development, as evidenced by differential gene expression, cell subpopulation analysis, and metabolic analysis. Additionally, our findings suggest that glial dysfunction, particularly in OPCs and Astro, plays a pivotal role in regulating the differentiation and migration of ExNs. Through cell-cell communication pathways, these glial cells influence the molecular environment, exacerbating neuronal migration defects. These results highlight the complex, multifaceted disruptions in cortical development in DS and provide a framework for understanding the molecular and cellular basis of cognitive and developmental impairments in individuals with DS during the mid-gestational stage.

The priority of regional maturation likely influences region-specific neuronal projections and interactions with subcortical regions, ensuring the integration of the neural network[13]. Previous comparisons of the human developmental PFC and visual cortex have offered brief insights into distinct molecular regulations and developmental pathways between the caudal and rostral regions of the cerebral cortex[56]. This study provides a significant transcriptomic framework for understanding the cell-type-specific developmental changes in DS, with a focus on the PFC and STP regions. Our results indicated significant transcriptomic differences between the PFC and STP in fetuses with DS. Notably, STP cells showed strong activation, with 90% of DEGs upregulated, whereas the PFC exhibited more heterogeneous responses, particularly in ExN subtypes (ExN L5/6 and ExN L3-5), which were highly susceptible to DS-related pathology. Furthermore, the STP was more affected by chromosomal abnormalities than the PFC, with 192 DEGs across at least 10 STP cell types, compared to only 30 DEGs in the PFC. These findings align with previous research highlighting region-specific vulnerabilities in DS, where amyloid accumulation in the neocortex regions appears before the subcortical areas as found in postmortem DS studies[57]. A particularly significant finding of this study is the persistent imbalance between InNs and ExNs in fetuses with DS. Such imbalances are associated with various neurological conditions, including cognitive deficits and developmental delays, which are hallmark features of DS[58]. These findings provide a comparative analysis of how different brain regions and cell types respond to DS at the transcriptomic level.

In fetuses with DS, the majority of chr21 DEGs were found in neurons, contrasting with findings in adult individuals with DS, where microglia and endothelial cells were more affected[14]. Notably, RUNX1 and APP were significantly altered in the PFC and STP, respectively. Individuals with DS have a high age-related prevalence of AD and lifelong accumulation of brain Aβ in part due to the triplication of APP on chr21[59]. Virtual knockout analyses indicated RUNX1 regulates genes involved in neuronal development, while APP impacts dendritic transport and neuroepithelial differentiation. Transient abnormal myelopoiesis (TAM) is a common complication in newborns with DS. Partial tandem duplications of RUNX1 on chr21 were found, specifically in myeloid leukemia-DS samples, presenting its essential role in DS leukemia progression[60]. In addition to brain tissue, we also analyzed the transcriptome of peripheral blood in individuals with DS. Our results found that chr21-related DEGs were more upregulated in peripheral blood than in the neocortex, suggesting that peripheral blood may be more vulnerable to the pathological effects of DS. Interestingly, SON (a gene on chr21) was a significantly changed DEG in both the neocortex and peripheral blood, further indicating its potential role in DS. De novo mutations in SON can affect critical genes like TUBG1 and HDAC6, which are involved in neuronal migration and cortical organization, leading to intellectual disability syndromes[31]. These findings suggest that peripheral blood could serve as a more accessible model for understanding DS at the molecular level and highlight the potential

roles of specific genes such as SON, APP, and RUNX1 in driving DS pathology systematically.

Neuronal migration is an essential step of brain development, allowing displacement of neurons from their germinal niches to their final integration site[61,62]. In DS, deficits in proliferation and migration are particularly evident in prenatal PFC[11]. Our results reveal significant disruptions in neuronal migration, particularly affecting ExN subtypes such as CUX2-enriched L2 and RORB-enriched L4 neurons. These subpopulations, critical for cortical organization, exhibited altered developmental trajectories in fetuses with DS, with more immature neurons and fewer mature neurons at GW23. Notably, CUX2-enriched L2 neurons, which are associated with neuropsychiatric disorders such as AD, MS, and SCZ, were disrupted, indicating a potential link between abnormal cortical development in DS and subsequent neurodevelopmental and psychiatric conditions. Previous studies observed that individuals with DS have a smaller hippocampus from GW19-20 and overall reduced brain size from GW23, suggesting an impaired neurogenesis and/or migration of late-born cells, especially GABAergic interneurons[63]. In our study, by analyzing the distribution of neurons at GW23, we observed that FOXP2+ deep-layer neurons and SATB2+ upper-layer neurons also displayed abnormal distributions, suggesting a failure in the inside-out cortical migration pattern typical of normal brain development in the PFC and STP of DS fetal brains. This finding aligns with previous reports and further supports the notion of impaired inside-out cortical migration in DS, which is a hallmark of typical cortical development[11,64]. The species-specific expression of FOXP2 also leads to species-specific transcriptional changes in the L4 ExN or microglia, including upregulation of IL1RAPL2, involved in dendritic spine formation, and DSCAM implicated in DS[65]. This abnormal migration pattern likely disrupts proper cortical layer formation and neuronal connectivity, which are essential for normal brain function[66]. Interestingly, the expression of neuron migration genes in normal individuals followed fetal-stage specificity, with distinct gene expression patterns observed at different developmental stages. However, in individuals with DS, most migration-related genes were overexpressed at GW23 and lacked the stage-specific, sustained expression observed in controls. This dysregulated gene expression may contribute to the observed disruptions in neuronal migration and abnormal cortical layer formation in DS.

The development of the complex architecture of the mammalian brain requires coordinated timing of proliferation, migration, layering, and differentiation of distinct neuronal populations[67]. These cellular processes appear to be heavily dependent on the function of highly specialized microtubules, primarily composed of tubulin, in the developing neurons[68,69]. Our study further used migration-related DEGs as feature genes to construct machine learning models for diagnosing DS, which exhibited high accuracy in classifying the developmental states of neurons in both PFC and STP regions. The top contributors in these models were genes such as TUBA1A, TUBB2B, and TUBB2A, which play critical roles in neuronal microtubule dynamics and axon formation[67]. Our immunofluorescence staining further confirmed that TUBB2B+NeuN+SATB2+ triple-positive cells in DS brains showed significant distribution anomalies, supporting the snRNA-seq findings. Additionally, the study performed a virtual knockout analysis, revealing that several DS-associated genes could regulate neuronal microtubule dynamics and axon development through tubulin signaling. Future studies may focus on the molecular pathways underlying these disruptions and the potential therapeutic targets for correcting these developmental defects in DS.

Post-translational modifications (PTMs) of tubulin subunits, enriched in specialized microtubule structures, regulate microtubule dynamics at these specific sites[70,71]. A post-translational modification named protein lactylation, targeting lysine residues in both histones and non-histone proteins[37]. A recent study pointed out that the catalyzed lactylation of α-tubulin in a concentration-dependent manner

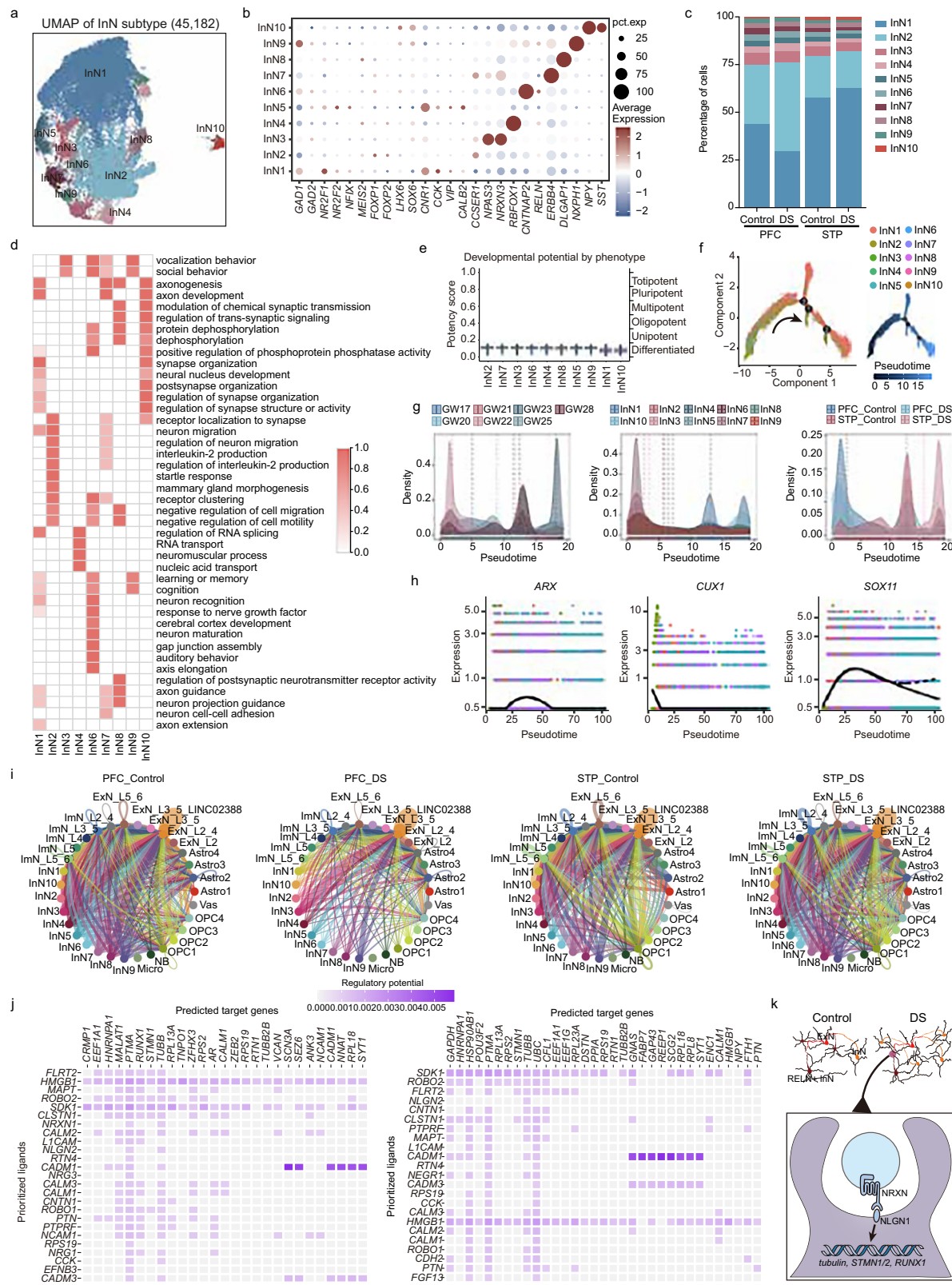

increases the dynamics of neuronal microtubules[39]. Our results revealed that the lactylation of α-tubulin, which regulates microtubule dynamics in neurons, has been disrupted in DS fatal brain. The abnormal lactylation of α-tubulin in neurons might contribute to altered neuronal structure and function by interfering with microtubule stability and neuronal connectivity. These findings underscore the complexity of the disease, where not only genetic factors but also

metabolic and epigenetic changes converge to disrupt normal brain development, leading to the cognitive and developmental challenges seen in individuals with DS. This research suggests that abnormal lactylation of α-tubulin may serve as a key driver of abnormal neuronal development and synaptic dysfunction in DS, offering potential avenues for therapeutic intervention aimed at correcting these metabolic imbalances to mitigate cognitive impairments.

**Fig. 6 | Subpopulation analysis and cell communication analysis of InNs.**
**a** Visualization of InN subclusters using UMAP. **b** Dot plot showing the average expression of the marker gene of InN subclusters. **c** Bar plot showing the proportions of InN subclusters in the control and DS from PFC and STP. **d** Heatmap showing the enriched GO terms of cell-type-specific marker genes of different InN subclusters with their enriched functional annotations on the right (FDR-adjusted *p*-value < 0.05). GO enrichment was performed using hypergeometric testing via the clusterProfiler package, and *p*-values were adjusted using the BH method. **e** Density box plot showing the developmental potential for different InN subclusters. The total number of cells analyzed was 29,857, with the number of cells selected per InN subcluster ranging from 559 to 10,000. Box plots indicate the median, interquartile range (25th–75th percentiles), and whiskers extend to 1.5 times IQR. **f** Pseudotime analysis of the InN cells. The points are colored by cell types (left) and pseudotime (right). The arrows indicate the directions of differentiation trajectories. **g** Density plot showing the distribution of gestational age, InN subclusters, and groups along the trajectory. **h** Pseudotime analysis showing the expression levels of the indicated transcription factor along the trajectory. **i** Network plot showing the weight of cell-cell interactions between indicated cell types in control and DS from PFC and STP. **j** Heatmap shows the potential regulation target genes of ExNs by ligands from InNs. **k** Heatmap shows the potential regulation target genes of ExNs by ligands from InNs. Source data are provided as a Source Data file.

Our study also delineated a crucial role of InNs and glial cells in the migration of ExNs during brain development in DS. The dysregulation of InNs, particularly the decrease in Reelin-expressing (*RELN*⁺) cells, has profound consequences on ExN migration and cortical development. Reelin is an essential regulator of neuronal migration[72], and its reduced expression, particularly in InN6 and InN9 subtypes, likely impairs proper neuronal positioning and cortical lamination. These developmental abnormalities contribute to cognitive dysfunction, as the migration of ExNs in regions like the PFC and STP is disrupted. Cellular communication between InNs and ExNs is significantly diminished in the PFC, particularly for key signaling interactions like *NRXN1-NLGN1*, *NRXN2-NLGN1*, and *NRXN3-NLGN1*; these interactions are enhanced in the STP. This regional variation suggests that while the PFC experiences a disruption in InN-ExN communication that likely exacerbates ExN migration defects, compensatory mechanisms may be at play in the STP, which could reflect regional adaptations to the developmental challenges in DS. The altered signaling between InNs and ExNs highlights the complexity of how cellular interactions influence neuronal migration and development in DS. Additionally, NicheNet analysis illuminated several critical signaling pathways in the regulation of ExN migration. Specifically, we found that InN-derived *NLGN2* and *PTN* signaling can influence ExNs by modulating the expression of key genes involved in neuronal migration, such as *RUNX1, STMN1, TUBB*, and *TUBB2B*. These molecules are integral to microtubule dynamics and neuronal movement, suggesting that the dysfunction of InNs in DS may lead to impaired ExN migration through disrupted regulation of cytoskeletal elements.

Alongside InNs, the dysfunction of glial cells, particularly OPCs and Astro, also plays a critical role in modulating ExN migration. OPCs are essential for supporting neuronal migration, differentiation, and maturation through both direct and indirect interactions with neurons[42]. The reduction of typical OPCs and immature oligodendrocytes observed in DS likely disrupts these essential roles, impeding the migration and maturation of ExNs. Astro, the most abundant glial cell type, are crucial for supporting and regulating ExN migration and synaptic development[73]. Furthermore, the observed increase in Astro3, a subtype expressing neural progenitor cell markers, suggests an abnormal differentiation of glial cells, which could further hinder the migration and differentiation of ExNs. The alterations in Astro populations—such as the reduced numbers of Astro2 and Astro4—indicate a disruption in intermediate progenitor cell functions, which are crucial for the proper formation and migration of neurons. These disruptions could affect synaptic plasticity and impair long-term cognitive and behavioral outcomes in DS. Notably, in the PFC, OPCs and Astro influence the expression of *RUNX1* in ExNs, which in turn modulates the expression of tubulin—a major structural protein essential for neuronal migration. In contrast, in the STP, these glial cells predominantly affect the expression of *DSCAM*, a molecule involved in neuronal differentiation and recognition. This regional specificity underscores the complexity of glial-ExN interactions and their differential roles in regulating ExN migration in different cortical areas. These findings highlight the intricate relationship between InNs, glial cells, and ExNs in DS. The dysregulation of both InNs and glial cells contributes to defects in ExN migration, which may underlie the cognitive and behavioral deficits observed in DS.

The disruptions in neuronal migration and glial-ExN interactions observed in DS share striking similarities with mechanisms implicated in other neurodevelopmental disorders, such as ASD and ADHD[74–76]. For example, *FOXP2* dysregulation has been linked to language deficits in ASD[77], while *TUBA1A* mutations are associated with cortical malformations in lissencephaly[78]. The proper development of the PFC relies on transcription factors, guidance cues, and other regulatory molecules, requiring a coordinated sequence of developmental processes. Any disruption in these processes can lead to neurodevelopmental abnormalities in the PFC, contributing to cognitive deficits seen in patients with neurodevelopmental disorders[75]. These parallels suggest that studying DS may provide a broader framework for understanding shared molecular and cellular pathways in neurodevelopmental diseases. Additionally, our findings highlight potential therapeutic targets, including lactylation regulators and tubulin-stabilizing agents, offering new avenues for intervention not only in DS but also in related conditions characterized by neuronal migration defects.

This study provides a comparative cellular and molecular atlas of cortical development in fetuses with DS, revealing dynamic changes in neuronal migration, glial cell function, and molecular signaling pathways. Future research should focus on refining DS models, such as cerebral organoids, to validate the functional roles of key genes like *RUNX1, APP, and tubulin* in neuronal migration. Additionally, exploring the interplay between epigenetic modifications, such as lactylation, and genetic factors could uncover novel mechanisms underlying neurodevelopmental disorders. By bridging the gap between transcriptomic findings and therapeutic applications, these efforts have the potential to mitigate the cognitive and developmental impairments in DS and related conditions.

We acknowledge several limitations in our study. First, the overall sample size is limited, with a small number of Ts21 fetal brain specimens available, and an imbalance in sex and gestational age across cases. These constraints stem from the exceptional rarity of obtaining mid-gestational fetal brain tissues with confirmed Ts21, which typically rely on elective terminations under strict ethical regulations and informed consent. As a result, achieving ideal matching across biological variables such as sex, gestational age, and region is practically unfeasible. Nevertheless, we emphasize that such rare samples provide a unique opportunity into the in vivo developmental biology of the Ts21 brain, offering insights that are otherwise inaccessible.

Another important limitation of our study lies in the potential sampling bias inherent to human fetal tissue research. The majority of DS samples included were derived from elective terminations following prenatal diagnosis, typically performed during mid-gestation. As such, these fetuses may represent a subset of Ts21 cases with comparatively milder genetic or cellular perturbations that were compatible with continued gestation to this stage. In contrast, more severely affected fetuses may be subject to earlier spontaneous loss and are therefore underrepresented. While this is a common constraint shared across similar studies, it nonetheless limits the generalizability of our findings to the broader DS population. We have attempted to address this issue

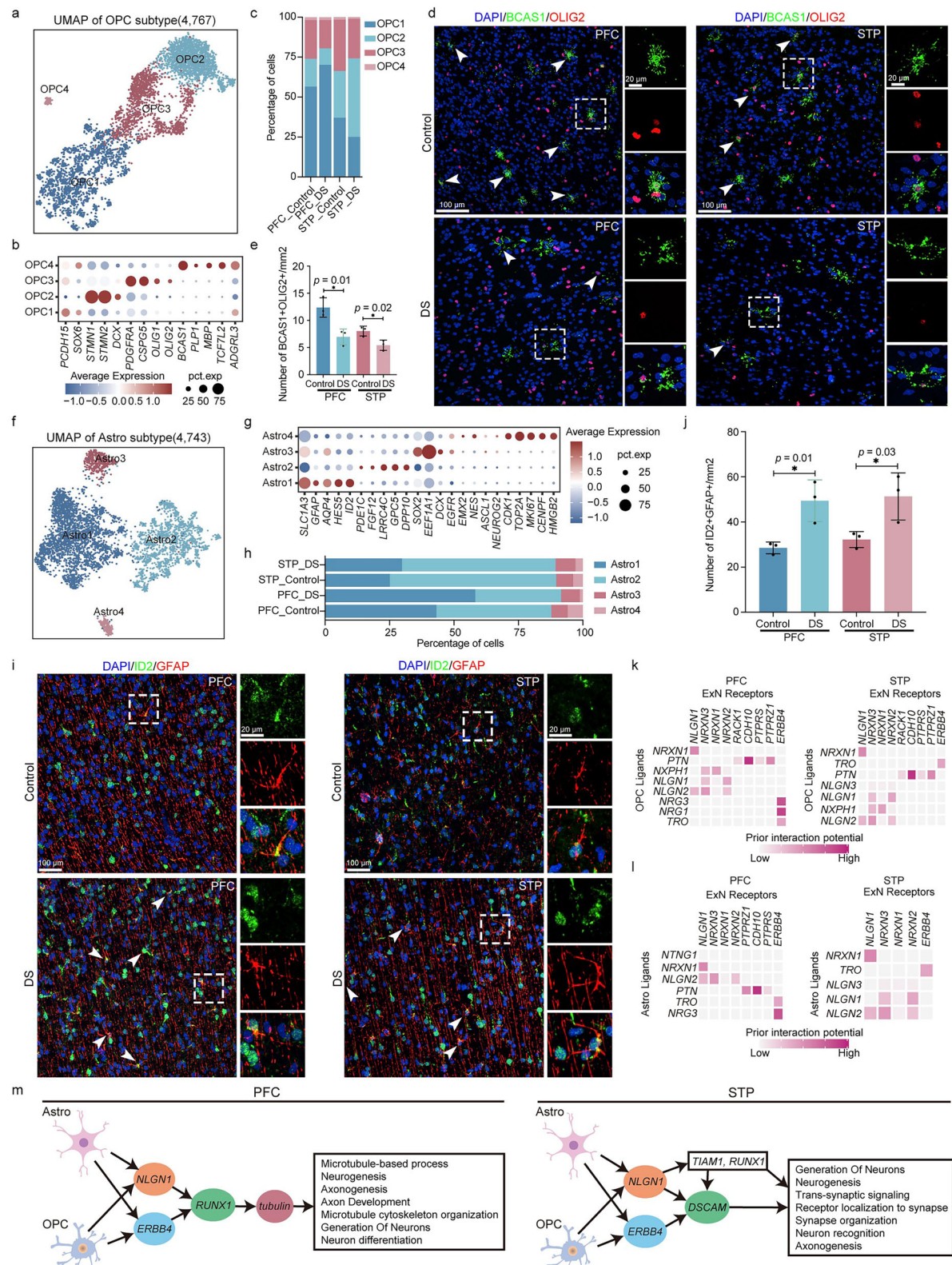

through stratified analyses and careful statistical controls, but future studies involving a broader range of gestational stages, ideally with integrated clinical outcome data, will be essential for fully capturing the developmental diversity and pathological spectrum of Ts21.

Despite the inherent limitations, we implemented several strategies to strengthen the clarity and interpretability of our findings. We included sex and gestational age as covariates in statistical modeling to minimize potential confounding effects. Furthermore, we performed exploratory stratified analyses where possible to assess the consistency and robustness of DEGs and cell-type-specific changes. We also carefully discussed potential selection bias, as surviving mid-gestational Ts21 fetuses may represent a relatively "healthier" subset. Together, these efforts allow us to more precisely delineate the molecular and cellular alterations associated with Ts21 during early cortical

**Fig. 7 | Subpopulation analysis and cell communication analysis of gliocyte.**
**a** Visualization of OPC subclusters using UMAP. **b** Dot plot showing the average expression of marker gene of OPC subclusters. **c** Bar plot showing the proportions of OPC subclusters in the control and DS from PFC and STP. **d** Representative microscopic fields of *BCAS1*(green)/*OLIG2* (red) positive cells in the PFC and STP from GW23 of control and DS. Blue, DAPI. Scale bar, 100 µm (low magnification), 20 µm (high magnification). White arrowheads indicate *BCAS1*/*OLIG2* positive cells, and white dashed boxes indicate the interest area. **e** Quantification of *BCAS1* (green)/*OLIG2* (red) double-positive cells in PFC and STP from Control and DS group. $n = 3$ biological replicates per group. Two-sided paired Student's *t*-test. Data are shown as mean ± SEM. Source data are provided as a Source Data file. *$p < 0.05$. PFC_DS vs Con: 95% confidence interval −10.32 to −1.541. STP_DS vs Con: 95% confidence interval −4.703 to −0.5140. **f** Visualization of Astro subclusters using UMAP. **g** Dot plot showing the average expression of the marker gene of Astro

subclusters. **h** Bar plot showing the proportions of Astro subclusters in the control and DS from PFC and STP. **i** Representative microscopic fields of *ID2* (green)/*GFAP* (red) positive cells in the PFC and STP from GW23 of control and DS. Blue, DAPI. Scale bar, 100 µm (low magnification), 20 µm (high magnification). White arrowheads indicate *ID2*/*GFAP* positive cells, and white dashed boxes indicate the interest area. **j** Quantification of *ID2* (green)/*GFAP* (red) double-positive cells in PFC and STP from Control and DS group. $n = 3$ biological replicates per group. Two-sided paired Student's *t*-test. Data are shown as mean ± SEM. Source data are provided as a Source Data file. *$p < 0.05$. PFC_DS vs Con: 95% confidence interval 8.357 to 37.39. STP_DS vs Con: 95% confidence interval 6.803 to 40.69. Heatmap shows the potential regulation target genes of the ExNs by ligands from OPC (**k**) and Astro (**l**). **m** OPC and Astro, respectively, regulate the development and positioning of neurons in PFC and STP through tubulin and *DSCAM*. Source data are provided as a Source Data file.

development. We view this work as an important exploratory step and a foundational resource that will inform and inspire more comprehensive studies with larger and more balanced cohorts in the future.

Collectively, this study investigated the cellular and molecular dynamics underlying cortical development in fetuses with DS, offering a detailed analysis of the altered mechanisms driving DS-associated neurodevelopmental abnormalities in the second trimester. By focusing on the PFC and STP, we identified region-specific disruptions in neuronal and glial cell populations, shedding light on the intricate regulatory networks governing cortical development in fetuses with DS at the mid-gestational stage. This work not only offers insights into potential molecular targets and cellular interactions but also establishes a robust foundation for future studies aimed at correcting neurodevelopmental abnormalities in DS.

## Methods
### Ethics statement
This study has been clinically registered (Trial registration: ChiCTR, ChiCTR2300070041), and human sample collection and research analysis were approved by the Affiliated Hospital of Zunyi Medical University (Approval No. KLL-2022-446). The informed consent was designed as recommended by the ISSCR guidelines for fetal tissue donation. Informed consent for fetal tissue procurement and research was obtained from the patient after her decision to legally terminate her pregnancy but before the abortive procedure. Participants in this study were not compensated for tissue donation; however, they received free diagnostic testing related to the experiment and were reimbursed for transportation costs incurred during participation in the study, in accordance with the ethical guidelines set by the Affiliated Hospital of Zunyi Medical University. Fetal brain tissue samples were collected after the donor signed an informed consent document that was in strict observance of the legal and institutional ethical regulations for elective pregnancy termination specimens at Affiliated Hospital of Zunyi Medical University. Ts21 was confirmed in all fetuses with DS through karyotype testing. All samples used in these studies had not been involved in any other procedures. All the protocols were in compliance with the 'Interim Measures for the Administration of Human Genetic Resources' administered by the Chinese Ministry of Science and Technology.

### Tissue sample collection and dissection
The study included 4 fetuses with DS (2 cases providing both PFC and STP samples, 1 case providing only PFC, and 1 case providing only STP), and 11 control fetuses (5 cases providing both PFC and STP samples, 4 cases providing only PFC, and 2 cases providing only STP) (Supplementary Data 1). Fetal brains were collected in ice-cold artificial cerebrospinal fluid containing 125.0 mM NaCl, 26.0 mM NaHCO$_3$, 2.5 mM KCl, 2.0 mM CaCl$_2$, 1.0 mM MgCl$_2$, 1.25 mM NaH$_2$PO$_4$ at a pH of 7.4 when oxygenated (95% O$_2$ and 5% CO$_2$). The fresh brain tissue was collected including 2 different regions: PFC and STP with 0.5 g pieces

for each region which was cut with a scalpel according to the human fetal brain map of Gray's Anatomy. The digestion was performed at 37 °C for 30 min in 20 mL digestion buffer containing 10 mg/mL collagenase IV, 15 µg/mL DNase I, followed by gently shaking once every 5 min. Subsequently, the digestion reaction was stopped by using Magnetic-Activated Cell Sorting (MACS) buffer and filtered with a 40 µm cell strainer. After centrifugation and resuspension, the cells were stained for flow cytometry analysis.

### Nucleus extraction for snRNA-seq
The approaches for nuclei extraction for snRNA-seq libraries generation in this study were the same as in a previous study[79]. For the nuclei extraction process, the tissues were thawed and cut into small pieces, then transferred into a homogenization buffer containing 20 mM Tris pH 8.0, 1 × protease inhibitor cocktail, 500 mM sucrose, 0.1% NP-40, 0.2 U/µL RNase inhibitor, 1% bovine serum albumin (BSA), and 0.1 mM DTT. The tissue pieces were ground 15 times with tight pestles and filtered with a 30 µm cell strainer. The supernatants were carefully discarded after being centrifuged at $500 \times g$ for 5 min at 4 °C. The pellets (nuclei) were resuspended in PBS containing 1% BSA and 20 U/µL RNase inhibitor for later snRNA-seq library construction.

### Single-nuclei library construction and sequencing
Briefly, single-cell suspensions were used to generate barcoded snRNA-seq libraries by droplet generation, emulsion breakage, bead collection, reverse transcription, as well as cDNA amplification. The mRNA capture was performed on a DNBelab C4 device (MGI Tech Co., Ltd.). cDNA amplification and library construction were generated using the DNBelab C series reagent kit (MGI Tech Co., Ltd.) following the manufacturer's instructions. All the libraries were sequenced on the DNBSEQ-T7 platform.

### Pre-processing and quality control of snRNA-seq data
Cell Ranger 7.0.1 (10x Genomics) was used to process the raw sequencing data. The sequencing data from snRNA-seq was filtered, and the gene expression matrix was obtained using DNBelab C Series scRNA analysis software. To further ensure the quality of our dataset, we only retained cells with the number of detected genes greater than 200 and the percentage of detected mitochondrial genes less than 5%. Doublets were detected using DoubletFinder (v.2.0.3). After sample integration and clustering, clusters lacking specific marker genes, with relatively low gene content and high mitochondrial ratios, were discarded.

### Identification of cell clusters
Briefly, UMIs from each valid cell barcode were retained for all downstream analyses and analyzed using the R package Seurat (v.4.2.2). After filtering, datasets from different sequencing libraries underwent normalization (using "NormalizeData()" function with parameters "normalization.method = "LogNormalize", scale.factor =

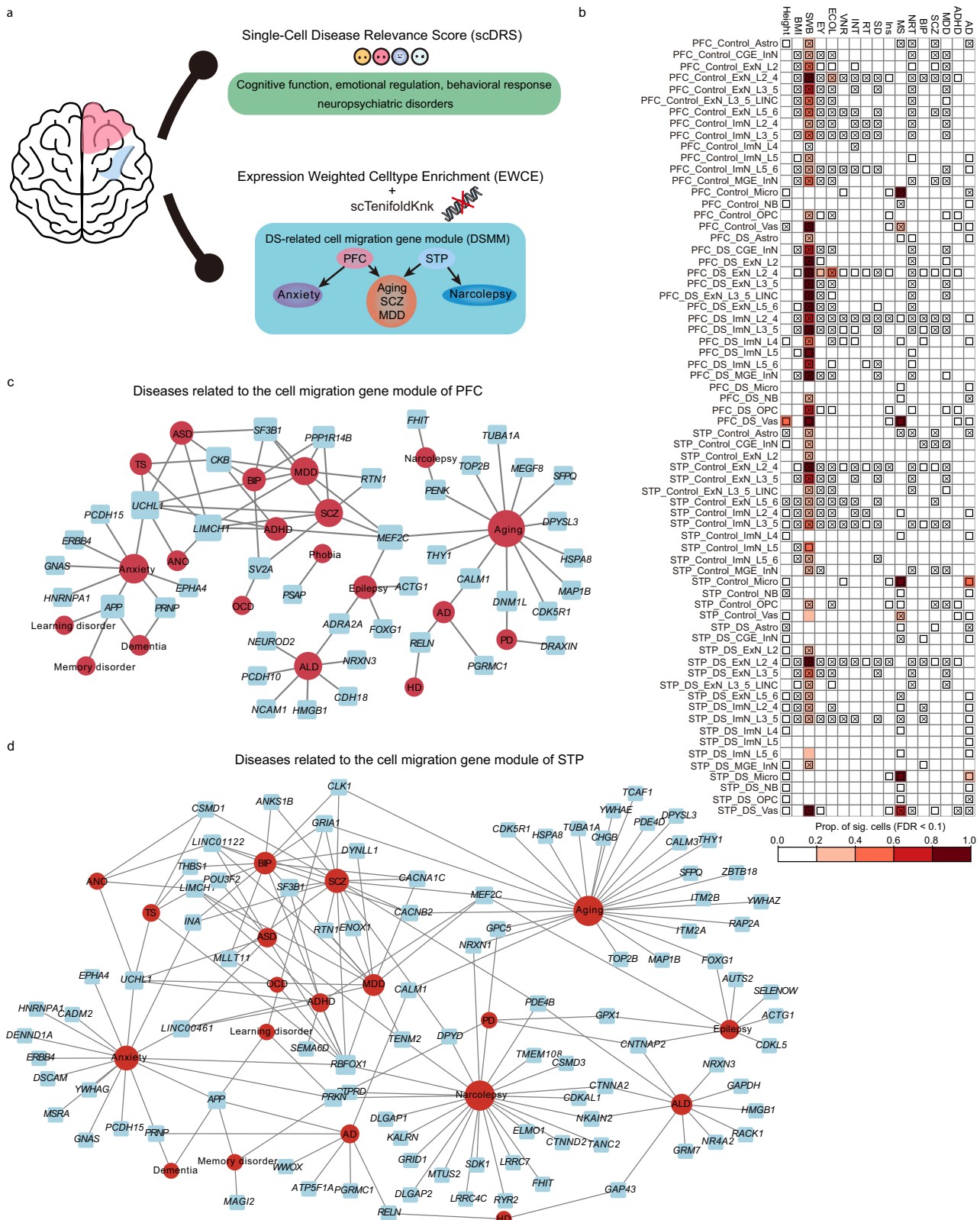

10,000") and identification of highly variable genes (HVGs) (using "FindVariableFeatures()" function with the options "selection.-method = "vst", nfeatures = 3000"). Then, we applied "FindInte-grationAnchors" and "IntegrateData" functions to integrate all the sequencing libraries with the top 30 significant principal components (PCs) (dim = 1:30). The top 3000 HVGs of each dataset were used for

downstream PC analysis (PCA). The top 30 significant PCs were selected for clustering and visualization using UMAP.

### Identification of cell-type-specific marker genes

"FindMarkers()" function implemented in Seurat v.4.2.2[80] was used to identify cell-type-specific marker genes with the options

**Fig. 8 | The enrichment differences of neuropsychiatric disorders and advanced brain functions in the PFC and STP of patients with DS. a** The framework for analyzing brain functions and diseases related to patients with DS includes two approaches. The first one is the trait/disease association analysis based on scDRS, which elucidates the changes in cognitive function, emotional regulation, and behavioral response in different cortices of DS. The second one is the disease association analysis of DSMM based on differential gene analysis and virtual KO construction. **b** Heatmaps display the proportion of significant cells (FDR < 0.1) associated with various neuropsychiatric disorders and advanced function across regions and groups. Squares denote significant cell–disease associations across all pairs of cells and diseases/traits. Cross symbols denote significant heterogeneity in association with disease across individual cells within a given cell type. Red shading indicates higher proportions of significant cells, with darker shades reflecting stronger associations. Diseases related to the cell migration gene module of PFC (**c**) and STP (**d**). Source data are provided as a Source Data file. AD Alzheimer's disease, ADHD attention deficit hyperactivity disorder, ANO anorexia nervosa, ASD autism spectrum disorder, FTD frontotemporal dementia, HD Huntington's dementia, MDD major depression disorder, PD Parkinson's disease, SASP senescence-associated secretory phenotype, TOS Tourette syndrome, SCZ schizophrenia, BIP bipolar disorder, MS multiple sclerosis, NRT neuroticism, INS insomnia, SD sleep duration, RT reaction time, INT intelligence, VNR verbal numeric reasoning, ECOL education college, EY education years, SWB subject wellbeing, BMI body mass index, ALD alcohol dependence, OCD obsessive–compulsive disorder.

"logfc.threshold = 0.25, min.pct = 0.25". *P*-value was corrected using the Bonferroni method, and 0.05 was set as a threshold to define significance.

### Gene ontology (GO) term enrichment analysis
The "enrichGO()" function of the clusterProfiler R package[81] was used for enrichment analysis, and the Benjamini-Hochberg (BH) method was employed for multiple test correction (OrgDb = org.Hs.eg.db, pAdjust-Method = "BH", pvalueCutoff = 0.05). A GO term with an adjusted *p*-value lower than 0.05 was considered significantly enriched.

### Cell composition analysis
To quantify the impact of aging on cell differentiation dynamics, we employed two distinct analytical methods for inter-group cell composition difference analysis. These included a cell composition analysis utilizing the Cacoa package[25] and a differential cell density analysis conducted with the miloR package[82]. The estimateCellLoadings() function from the Cacoa package (v.0.4.0) was utilized for analyzing cellular components. Specifically, logarithmic ratio transformation was applied to the fractions of different cell types, followed by typical discriminant analysis using the candisc software package to derive weighted comparisons between the two sample groups. The separation coefficient was assessed through random sub-sampling of cells, and its robustness, along with statistical significance, was thoroughly evaluated. Then, we perform 1000 resamplings, during which 1000 cells are randomly selected from each group to assess the robustness of the test. BH procedure is employed for multiple comparison correction. Cacoa package's estimateCellDensity function is utilized to conduct a differential cell density analysis. To assess the differences in cell density between sample groups, we first employ the ks R package to compute the kernel density for each sample within the joint embedding space. Subsequently, we normalize the resulting density matrix across samples using quantile normalization techniques. To quantify the differences in cell density between sample groups, we perform a *t*-test on samples located within each grid box. To mitigate background noise, we filtered out boxes containing at least one cell, and the z-scores were represented as a heatmap. The sample labels were randomly shuffled 200 times to assess the robustness of the test. For single-cell differential abundance analysis, we utilized the Milo function to create a miloR object, followed by employing the build-Graph function to construct a K-nearest neighbors (KNN) graph in UMAP space. Subsequently, the makeNhoods function was applied to define cellular neighborhoods, while the countCells function quantified the number of cells within each neighborhood across all samples. The testNhood function was employed to evaluate neighborhood differential abundance with a spatial false discovery rate (FDR) significance threshold set at 0.05. Visualization of differential abundance neighborhoods was achieved using the plotNhoodGraphDA function.

### Pseudotime trajectory analysis
Monocle2 package was used to discover the cell state transitions. Genes expressed in less than 5 cells were filtered out. DEGs were computed by the function "differentialGeneTest" in monocle2. Genes with a qvalue less than 0.01 were regarded as DEGs and sorted by qvalue using the "setOrderingFilter" function. The pseudotime trajectory was constructed by the "DDRTree" algorithm with default parameters. The dynamical expression changes of selected marker genes by pseudotime were visualized by the "plot_genes_in_pseudotime" and "plot_pseudotime_heatmap" functions. CytoTRACE analysis was carried out using the CytoTRACE R package (v.0.3.3). Following the generation of the gene-cell expression matrix, the trajectory was then inferred using default parameters and visualized using the "plotCyto-TRACE" function.

### Disease prediction of cells using machine learning
The snRNA-seq data of ExNs or InNs were selected for disease classification. We selected DEGs or genes associated with cell migration as input and disease state as output for machine learning. The pipeline and functions were implemented in the automated machine learning R package h2o (v.3.44.0.2). For data splitting, 25% of nuclei were first split into the testing set, and the rest 75% were further split into training and validation sets using 10-fold cross-validation and 20 max models. The best model was extracted using the function "h2o.get_best_model", evaluated on the test and validation sets with "h2o.predict", and the permutation-based variable importance was obtained using "h2o.permutation_importance_plot".

### Gene set score analysis
Gene set scores were acquired by analyzing the transcriptome of each input cell against the aforementioned gene sets by the Seurat function "AddModuleScore". Changes in the scores between groups were analyzed using the ggpubr R package via the Wilcoxon test.

### Expression-weighted cell-type enrichment analysis
Expression-weighted cell-type enrichment (EWCE)[79] was employed for the analysis of disease gene enrichment using default parameters. This analysis was conducted separately for each species dataset. This analysis was performed separately for each species dataset. To mitigate the bias effect, we initially employ the "vst()" function to conduct a variance-stabilizing transformation on the unique molecular identifier count matrix. Subsequently, we utilize the "fix_bad_mgi_symbols()" function to rectify gene nomenclature. The "drop_uninformative_genes()" function is then applied to eliminate non-informative genes, thereby reducing computational time and minimizing noise in subsequent analyses. Following this, we implement the "generate.celltype.data()" function to compute the specificity matrix for each dataset. We perform 100 bootstrap resamplings with replacement on all detected genes within each species dataset as a background reference, and *p*-values are adjusted using the BH method. A significance threshold of 0.05 is established.

### Gene-regulatory network
To identify organ-specific gene-regulatory networks, we performed Single-cell Regulatory Network Inference and Clustering (v.0.12.1; a

Python implementation of PySCENIC)[83]. Firstly, data was subsampled by randomly selecting 800 cells from each tissue/organ. The original expression data were normalized and multiplied by 10,000, followed by a log1p transformation. Next, normalized counts were used to generate the co-expression module with the GRNboost2 algorithm implemented in the arboreto package (v.0.1.3). Finally, we used pyS-CENIC with its default parameters to infer co-expression modules using the above-created RcisTarget database. An AUCell value matrix was generated to represent the activity of regulators in each cell. GRNs were visualized by the igraph package in R and Cytoscape (v.3.9.0).

## Single-cell disease relevance score (scDRS)

ScDRS was used to evaluate polygenic disease enrichment of individual cells in snRNA-seq data[84]. GWAS summary statistics for 17 diseases/characteristics (average $N = 346$ K) are downloaded directly from the GitHub page of scDRS. First, to determine statistical significance, scDRS generates 1000 sets of cell-specific raw control scores at Monte Carlo samples of matched control gene sets (matching gene set size, mean expression, and expression variance of the putative disease genes). Next, scDRS normalizes the raw disease score and raw control scores for each cell (producing the normalized disease score and normalized control scores), and then computes cell-level $p$-values based on the empirical distribution of the pooled normalized control scores across all control gene sets and all cells.

## Identification of DEGs

To identify DEGs in aging or sex, $p$-values were calculated and FDR-corrected using Model-based Analysis of Single-cell Transcriptomics (MAST)[33]. All nuclei from different group samples for corresponding cell types were used. MAST was used to perform zero-inflated regression analysis by fitting a linear mixed model. To exclude gene expression changes stemming from confounders, such as age, sex, fractions of ribosomal and mitochondrial transcripts, the following model for aging and sex was fit with MAST:

$$\text{zlm}( \sim \text{condition} + \text{nCount\_RNA} + \text{percent.mt} + \text{Sex, sca, method} = \text{glmer, ebayes} = T)$$

$$\text{zlm}( \sim \text{condition} + \text{nCount\_RNA} + \text{percent.mt} + \text{Age, sca, method} = \text{glmer, ebayes} = T)$$

Where percent.mt is mitochondrial RNA fraction.

To identify genes differentially expressed due to the age effect, a likelihood ratio test was performed by comparing the model with and without the diagnosis factor. Genes with at least 25% increase or decrease in expression in a group versus another group and a FDR-corrected $\underline{p}$ < 0.05 were selected as differentially expressed.

## Virtual knockout of the gene of interest

To elucidate the effect of DEGs knockout on cell-specific function, we extracted the snRNA-seq data of the cell type and used the expression matrix of genes × cells as the input for scTenifoldKnk[85]. The virtual knockout perturbed genes with FDR-corrected $p$ < 0.05 were selected as differentially expressed. The interaction enrichment analysis was provided and based on the STRING protein-protein interaction database. The R package Enrichr (v.3.2) was used for functional enrichment analysis. In the scTenifoldKnk analysis, the perturbation effect of a virtual gene knockout was quantified by computing the Euclidean distance between the network embeddings of each gene before and after the knockout. To assess statistical significance, $p$-values were computed using a Chi-square distribution with one degree of freedom, based on the fold change relative to expected values. These $p$-values were subsequently adjusted for multiple testing using the BH method.

## Integrated data source

To provide a developmental and systemic context for our fetal cortical findings, we incorporated three orthogonal datasets with proper attribution. Age-dependent comparisons utilized published PFC data from a previous study on DS brain aging[14]. This cohort included 23 postmortem samples: Aged group (56–65 years): 6 neurotypical controls, 4 individuals with DS; Young group (12–39 years): 8 neurotypical controls, 5 individuals with DS. Reference transcriptional trajectories during in vitro corticogenesis were obtained from Tang et al. [29], encompassing single-cell RNA-seq of iPSC-derived cortical organoids across differentiation stages. Peripheral blood transcriptome analyses utilized published data from Waugh et al. [30], comprising whole-blood RNA-seq from 304 individuals with DS and 96 euploid controls. All external datasets underwent identical normalization and batch correction procedures as our primary fetal snRNA-seq data.

## Data stratification and cross-validation

The collected samples were concentrated in the mid-gestational period. Given the extreme rarity of such specimens, we initially prioritized using all available samples for global analyses while controlling for potential confounding factors such as sex and gestational age where possible. However, due to the broad gestational range and sex imbalance between groups, we also performed stratified analyses based on gestational age and sex. For gestational age, we generated three additional stratified datasets: GW21–23, GW21–28, and GW21–28 matched. The GW21–28 dataset consisted of PFC samples matched by propensity scores, and the GW21–28 matched dataset included both PFC and STP samples following propensity score matching.

For sex-based stratification, due to the limited number of male fetuses with DS ($n = 1$), we performed an independent differential expression analysis focusing on female samples ($n = 2$ DS vs. 4 controls), where sufficient statistical power could be achieved. Although formal testing on the male sample was not feasible, we conducted an exploratory evaluation of expression patterns in the male fetus with DS.

We employed the following formula to calculate the consistency between stratified and global results:

$$\text{Consistency Rate}(\%) = \frac{N_{\text{consistent}}}{N_{\text{total}}} \times 100 \qquad (1)$$

$N_{\text{consistent}}$: The number of items showing consistent results between global and stratified analyses. In cell proportion analysis, $N_{\text{consistent}}$ is the number of cell types whose direction of change (increase or decrease) is the same in both global and stratified analyses. In DGE analysis, it is the number of genes that are significantly differentially expressed (e.g., adjusted $p$-value < 0.05) in both analyses and share the same direction of change (e.g., both upregulated or downregulated). In gene set enrichment analysis (e.g., chr21 genes), it refers to gene sets that are significantly enriched in both analyses with consistent enrichment direction. In EWCE disease enrichment, $N_{\text{consistent}}$ represents the number of cell types showing significant enrichment and the same direction of association (positive or negative Z-score) with the disease gene set in both analyses. $N_{\text{total}}$: The total number of items compared (e.g., cell types, genes, gene sets, or disease associations).

The interpretation framework was defined: High consistency (>70%) indicates disease susceptibility and generality of the gene set across subgroups. Moderate consistency (50–70%) suggests partial confounding effects, with some genes potentially influenced by sex or gestational age. Low consistency (<50%) implies strong subgroup-specific effects, likely reflecting significant gestational age or sex-related variability. Additionally, to validate the robustness and biological relevance of our findings, we performed cross-validation with

previously published datasets, including GSE168408, GSE204684, and GSE59630.

## Multiplexed immunofluorescence staining

Brain tissues of the GW23 fetus from the control and DS were used for multiplexed immunofluorescence staining. The cortical tissues were dissected and fixed with 4% paraformaldehyde (BL539A, Biosharp) for up to 24 h and sequentially placed in 10%, 20%, and 30% sucrose (21164955, Biosharp) at 4 °C until the tissues were completely immersed in each solution. The tissue samples were frozen with optimal cutting temperature compound (4583, Tissue-Tek) at −40 °C for 30 min and sectioned at a thickness of 15 μm using a microtome (CM1950, Leica). Sections were then stored at −40 °C. After antigen repair with citrate-EDTA antigen retrieval solution (1×) (BL55A, Biosharp), the sections were incubated with 3% hydrogen peroxide (Jiangxi Grass Coral Disinfection Products Co. Ltd.) for 15 min and blocked with 5% goat serum (SL038, Solarbio) and 0.3% Triton X-100 (1139ML100, BioFROX) for 2 h. Subsequently, the sections were supplemented with primary antibodies for 18 h at 4 °C. This was followed by rinsing in PBS with Tween 20 (PBST) 5 times (5 min each). Secondary antibodies (kit-5020, MaxVision-HRP, mouse/rabbit) were incubated at room temperature for 15 min. This was followed by washing with PBST and incubation for 10 min with TSAPlus fluorescent enhancement dye (iF488-Tyramide, iF555-Tyramide, or iF647-Tyramide) (GB1236, Servicebio). For double/triple-label staining, these sections were subjected to antigen repair again with citrate-EDTA antigen retrieval solution and were incubated with 3% hydrogen peroxide for 15 min, followed by blocking with 5% goat serum and 0.3% Triton X-100. Incubation with primary and secondary antibodies was performed. Detailed primary antibody information is provided in Supplementary Data 19. Finally, after DAPI staining for 10 min, sections were sealed with an anti-fluorescent quenching agent (Po126, Beyotime). Fluorescence images were captured with a Nikon AX confocal microscope and NIS-Elements AX software (image size 2048 × 2048 pixels).

Cortices were immunostained for *FOXP2*, *SATB2*, *TUBB2B*, and *NEUN*, and random cortical structures were imaged on a confocal microscope. From the pial surface, radial columns covering the full thickness of the cortical plate (CP) were cropped for analyses. The positions of all positive nuclei were separately marked using the "Cell Counter" plugin in ImageJ. Their y-coordinates on the image were recorded and normalized to the full thickness of the CP to measure their relative laminar positions in the CP. The frequency distributions of the relative vertical positions in 10 evenly divided bins for each marker were calculated and plotted in Prism software (GraphPad). The relative locations of all marker+ nuclei were used to plot the cumulative distribution curve and calculate the P-values.

## Statistical and reproducibility

If not specified, all statistical analyses and data visualization were done in R (v.4.2.2). The number of replicates and statistical analysis results are detailed in the corresponding figure legends. Data from multiplexed immunofluorescence staining were collected from at least 3 independent experiments and presented as the mean ± SEM. Statistical analysis was performed using Prism 10.3.1 software (GraphPad).

## Reporting summary

Further information on research design is available in the Nature Portfolio Reporting Summary linked to this article.

## Data availability

The raw sequence data from our study have been deposited in the Genome Sequence Archive for Human in the National Genomics Data Center under the accession code HRA012374 (https://ngdc.cncb.ac.cn/gsa-human/browse/HRA012374). All original data generated or analyzed in this study are available in the Supplementary Information and

Source Data file. Source Data are provided with this paper. Previously published datasets GSE168408, GSE204684, and GSE59630 were used for integrated analysis. Source data are provided with this paper.

## Code availability

This paper does not report original code, and publicly available tools were used in data analysis. All codes used in this study are available at https://github.com/niuruize/Cell-Type-Specific-Transcriptomic-Signatures-in-the-PFC-and-STP-Cortex-of-DS-Fetuses.

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

## Acknowledgements

We sincerely thank You Wu, Ye-Qing Fei, and Cheng-Lin Song from the Department of Family Planning, The Affiliated Hospital of Zunyi Medical University, for their assistance in obtaining the clinical samples used in this study. We would also like to sincerely thank Beijing Genomics Institute (BGI) for their valuable technical guidance on sequencing data analysis. This study was financially supported by Guizhou Provincial Higher Education Science and Technological Innovation Team (grant number: [2023]072, L.L. Xiong), Guizhou Province Distinguished Young Scientific and Technological Talent Program (grant number: YQK[2023] 040, L.L. Xiong), National Natural Science Foundation of China (grant number: 82560317, L.L. Xiong), Zunyi Medical University 12345 Future Talent Training Program–Technology Elite (grant number: ZYSE-2021-03, L.L. Xiong), and the Talent Research Startup Fund from the First People's Hospital of Zunyi (L.L. Xiong).

## Author contributions

L.L.Xiong, T.H.W. and C.L. conceptualized and designed the study and revised the manuscript. R.Z.N. contributed to snRNA-seq data analysis and writing of the "Results" section. L.L.Xue and X.H.T. were responsible for clinical sample ethics, registration, sample collection, processing and sequencing. L.R.H. carried out immunostaining identification. LC drafted the manuscript and edited the revision. S.F.W., Y.Y.Z., H.Y.Q. and Z.J.G. performed histochemistry quantification. Y.Y.Z. and C.Y.Z. collected clinical samples.

## Competing interests

The authors declare no competing interests.
