## [Transparent Peer Review file · Nature Communications]

Mid-gestational cell-type specific transcriptomic signatures in the prefrontal and superior temporal cortex in Down syndrome

Corresponding Author: Professor Liu-Lin Xiong

Version 0:

Reviewer comments:

Reviewer #1

(Remarks to the Author)

The study by Niu et al. is aimed at detecting possible gene expression abnormalities in the developing cortex of fetuses with Down syndrome (DS). To this end, they carried out single-nucleus RNA sequencing using samples from the prefrontal cortex and superior temporal plane of four fetuses with Down syndrome and eleven control fetuses, aged 17-28 GW. Gene expression was analyzed in each cell type (e.g., excitatory neurons, inhibitory neurons, glia) populating the developing cortex which were recognized based on their transcriptome. The study shows cell-specific and region-specific gene expression changes in fetuses with DS vs. controls. In addition, the study reports a relative increase in the number of inhibitory vs. excitatory neurons and abnormalities in their cortical distribution in fetuses with DS vs. controls. These abnormalities appear to change across different gestational times. The study is well designed and provides very useful information on gene expression abnormalities during cortical development in DS, a field scarcely explored due to the limited availability of fetal material. The article is altogether well-written and well-discussed. I have some suggestions that the authors may wish to address.

Line 2. The title is too ambitious because the study is not "a comprehensive transcription atlas" of cortical development, being focused on two brain regions only. I suggest changing the title in a more realistic manner.

Lines 24 and 28. Here, as well as in other parts of the text (lines 94, 119, 122, 136, 241, 242), the authors use the term "embryonic" referring to the brains they have analyzed in this study. In humans the embryonic period finishes at GW8. Thus, the term "embryonic" should be replaced with "fetal".

Line 34. The expression "comprehensive spatiotemporal analysis" is too ambitious because the gene expression analysis is limited to two cortical regions and because in the case of fetuses with DS the number of cases per gestational age is very small (2 fetuses at GW21, 1 fetus at GW23 and 1 fetus at GW28).

Line 56. "This process is particularly pronounced". It is not clear what is meant here with "process". If it does refer to corticogenesis I don't think that corticogenesis is more pronounced in the regions examined in this study.

Lines 89 and 89. The statement "with groups matched per GW, gender and postmortem interval" is misleading. The brain samples of fetuses with DS derive from 3 females and 1 male only, so I do not understand the statement "gender matched". Regarding gestational ages, the samples of fetuses with DS consist in 2 cases at GW21, 1 case at GW23, and 1 case at GW28. Thus, I do not understand the comparisons that are reported in Supplementary Fig. 1A. Although it is not specified, it seems that the first graph from left (gestational age) refers to the samples used for the analysis of the prefrontal cortex and the second graph refers to the cases used for the analysis of the superior temporal plane. In fact, there are 3 data points on the column labeled DS and 3 data points on the column labeled DS in the first and second graph from left, respectively. The same holds for the third and fourth graphs from left (postmortem interval). If so, this is not correct. The comparisons of age and postmortem intervals should include all the fetuses with DS (n=4) and controls (n=11).

Lines 98-99. Regarding cell taxonomy and nomenclature of cell types, it is important to specify the studies from which they are derived.

Lines 127-128. The statement “leading to consistent overexpression of chr21 related genes” is not completely true because not all the genes on HSA21 are over expressed (Antonarakis SE et al.. Nat Rev Dis Primers. 2020 Feb 6;6(1):9). Indeed, an analysis of DS lymphoblastoid cells showed that only 29% of HSA21 genes are sensitive to the gene-dosage effect (expression close to the expected value of 1.5) or are amplified (expression >1.5), 56% are compensated (expression <1.5), and 15% are highly variable among individuals (Ait Yahya-Graison E, et al. Am J Hum Genet. 2007 Sep;81(3):475-91). Moreover, the sentence “analysis of 621 chr21 genes revealed elevated expression across all cell types in DS fetuses compared to controls” is not clear. It seems that all the 621 analyzed genes were overexpressed in fetuses with DS, which is unlikely.

Line 131 “DS fetuses” The authors should use person-first language here and throughout their report (e.g., “fetuses with DS” instead of “DS fetuses” or “individuals with DS” instead of “DS individuals”).

Lines 186-187. “To elucidate age-dependent transcriptional changes we compared the PFC of young and aged DS individuals (Supplementary Table 8)”. In the Methods, however, there is no mention regarding these cases (young and adults). Perhaps these data derive from cases used in previous studies? If so, this should be specified. Please note that data of young and adult cases are reported in a Table that is headed as “Supplementary Table 7” and not as “Supplementary Table 8”.

Line 195. Again, I suppose that the data regarding organoids reported in Fig. 2 derive from a previous study. Which study? This point should be clarified.

Line 210. “Transcriptomic analysis of peripheral blood...”. Same comment as above,

Lines 238-256. The analysis of changes in the abundance of certain subpopulations of neurons in the examined regions across ages has a major flaw because in the case of fetuses with DS it is based on 2 cases at GW21, 1 case at GW23, and 1 case at GW28. Moreover, while the cases at GW21 and GW23 were females, the case at GW28 was a male. Thus, interpretation of results across ages seems hazardous. The same holds for the data presented in the following part of the section “Abnormal alterations of ExN subtypes in the neocortex of DS fetal brain”. It is true that it is difficult to get access to fetal samples, but data needs to be strengthened by more cases to be convincing. Thus, I suggest shortening the whole section and interpreting results. with more caution.

Lines 445-446. The sentence “The differential changes....” needs to be clarified.

Line 487. “The findings underline the higher disease susceptibility of the STP region...”. Citations that this region is more susceptible to damage should be provided.

Minor

Line 52. “... abnormalities observed in DS is the disruption of cortical development, fundamentally alters...), I think that a “that” is missing between “development” and “alters”.-

Line 64. “Neurogenesis”. It should be specified that it is “cortical neurogenesis” that is characterized by an inward-outward pattern.

Line 86. Quotation [16] seems to be wrong here.

Line 118. “huan cortex” should be “human cortex”.-

Line 124. “two neocortex” should be “two neocortices”.-

Line 195. “The study” should be “A study”.

Line 237. “Abnormal alterations...”. An alteration is abnormal, by definition. Please reword the title,

Line 244. “Supplementary Figure 1D does nor report data of neurons at different differentiation stages.

Line 307. The cases are not 4 per group in the case of fetuses with DS, since 3 of the 4 cases were used for the PFC and 3 of the 4 cases were used for the temporal plane cortex.

Line 433. “of DS” should be “of fetuses with DS”.

Lines 482-483. Please specify the meaning of the acronyms.

Line 736. “FC” should be defined.

Line 741. The hippocampus has not been used in this study,

Line 744. "were followed" should be "were the same as in...".

Reviewer #2

(Remarks to the Author)

Reviewer #3

(Remarks to the Author)

The manuscript by Niu et al "A Comprehensive Transcriptomic Atlas of Cortical Development in Down Syndrome Fetuses" examines differences between development in midgestational fetuses with and without Trisomy 21 (Ts21). This work performed single nucleus RNA sequencing (snRNAseq) on cells from the prefrontal cortex (PFC) and superior temporal plane cortex (STP). For the PFC, 3 (2F 1M) and 9 (5M and 4F) fetuses with and without Ts21 were used; and for the STP 3 (2F 1M) and 7 (4M 3 F) fetuses with and without Ts21 were used. From these samples, this study purports to find alterations inhibitory-to-excitatory neuron ratio, dysregulated gene expression, differential cell development and distribution, and cellular pathways that are disrupted in these tissues in those fetuses with Ts21. Because Trisomy 21 is such a heterogeneous disorder with a wide variability in manifestation, it is difficult to believe the results presented from such a small number of fetuses, over a wide developmental stage, with no potential sex differences considered. There may be differences as outlined in the manuscript, but these are likely not based on enough samples to have the power to detect real differences between fetuses with and without Ts21.

There are only 3 fetuses with Ts21 used for sampling the STP and PFC. The gestational age for the fetuses with Ts21 were 21-28 weeks for STP and PFC, and without Ts21 were 17-25 weeks. Even though there were not statistically significant differences between the age ranges of the groups (small sample sizes hinder detecting such differences!), 1) there is a wide age range between fetuses of the study where a lot of temporal fetal developmental differences are found in brain tissue, and 2) the developmental age range of fetuses with and without Ts21 is different (not age matched as indicated in the manuscript). Furthermore, there is a wide range of differences between developing embryos and fetuses with Ts21, with up to 75% and 50% of fetuses spontaneously lost in the first and second trimesters, respectively. These data indicate a wide range of developmental differences in fetuses with Ts21; for example, 1 or more of the 3 fetuses with Ts21 sampled in this study may have had so much genetic or cellular dysregulation that it would have been spontaneously lost before the third trimester (if allowed to continue development), and the results given would not relate to data mentioned in the manuscript comparing them to live born individuals with Ts21. Thus, due to the high variability in development associated with Ts21, the sample sizes are likely not large enough and occur at too wide of a developmental age to detect correct potential differences for Ts21 development and support the conclusions of the manuscript.

The manuscript (supplemental information) lists the sex of each fetus, but no attempt is made in this study to account for sex specific differences in fetal development that would lead to sex differences in cognitive development that are becoming more recognized in individuals with Down syndrome (DS). Furthermore, there is only 1 male fetus with Ts21 at one gestational age (28 W) that makes it hard to believe any extension of these data to other male fetuses with Ts21. Based on these points, and the points made in the previous paragraph, these data do not seem generalizable to developing individuals with Ts21 in the ways that have been written in the manuscript. Additional data or analyses of some of these data in different ways would need to be done to make the results seem plausible, support the conclusions, and applicable to past and future and decrease the significance of these results.

Other concerns:

Please use person first language when referring to individuals with DS. Individuals with DS do not like to be referred to as "patients" and this reviewer would argue that a fetus does not have DS, but only exhibits trisomy 21. Therefore, it is person with DS, not DS patients or DS fetuses, and likely should be fetuses with Ts21 throughout the manuscript. The phrase "due to individual choice or other objective reasons" (line 47) seems offensive to parents with children with DS and should be eliminated.

Version 1:

Reviewer comments:

Reviewer #1

(Remarks to the Author)

The authors have addressed my concerns in an appropriate manner. I have noticed that expressions such as "DS fetus/es" and "DS patient/s" still appear in the revised version. Please replace with "Fetus/es with DS" and "people with DS", respectively. In addition, there are some sentences that need to be amended. For instance:

Lines 305-306. Please add a quotation regarding the role of InN6 and InN9.
Line 123. "...show greatly consist...". I suppose that the authors mean "show great consistency".
Lines 129-129. Something is missing in this sentence.
Lines 314-315. Something is missing in this sentence.
Lines 371-372. The sentence would be clearer by insertion of "risk genes" after "(MS)" and "(PD)", respectively.
Lines 377-378. Something is missing in this sentence.
Lines 425-428. The phrasing of these two sentences is somewhat clumsy. Please, reword them.
Line 472. "...the PFC and STP of DS fetal" I suppose that the word "brain" is missing after "fetal".
Line 488. What is meant with the expression "neuron states"?
Line 576 "Irreplaceable window". I think that "the rare cases" provide a "unique opportunity" rather than representing a window.
Line 634. "...including 3 different regions: PFC and STP". I suppose that "3" should be replaced by "2".

Reviewer #2

(Remarks to the Author)

Reviewer #3

(Remarks to the Author)

The authors have addressed my comments and concerns. I have no further questions or comments for the authors.

In cases where reviewers are anonymous, credit should be given to 'Anonymous Referee' and the source. The images or other third party material in this Peer Review File are included in the article's Creative Commons license, unless indicated otherwise in a credit line to the material. If material is not included in the article's Creative Commons license and your intended use is not permitted by statutory regulation or exceeds the permitted use, you will need to obtain permission directly from the copyright holder.

REVIEWER COMMENTS

Reviewer #1 (Remarks to the Author):

The study by Niu et al. is aimed at detecting possible gene expression abnormalities in the developing cortex of fetuses with Down syndrome (DS). To this end, they carried out single-nucleus RNA sequencing using samples from the prefrontal cortex and superior temporal plane of four fetuses with Down syndrome and eleven control fetuses, aged 17-28 GW. Gene expression was analyzed in each cell type (e.g., excitatory neurons, inhibitory neurons, glia) populating the developing cortex which were recognized based on their transcriptome. The study shows cell-specific and region-specific gene expression changes in fetuses with DS vs. controls. In addition, the study reports a relative increase in the number of inhibitory vs. excitatory neurons and abnormalities in their cortical distribution in fetuses with DS vs. controls. These abnormalities appear to change across different gestational times. The study is well designed and provides very useful information on gene expression abnormalities during cortical development in DS, a field scarcely explored due to the limited availability of fetal material. The article is altogether well-written and well-discussed. I have some suggestions that the authors may wish to address.

Q1: Line 2. The title is too ambitious because the study is not “a comprehensive transcription atlas” of cortical development, being focused on two brain regions only. I suggest changing the title in a more realistic manner.

Response: We sincerely appreciate the reviewer’s constructive suggestion. We agreed that the original title overstated the scope of our study, as our analysis was indeed restricted to two cortical regions (prefrontal cortex and superior temporal plane). To better reflect the focus of our work, we have revised the title to: "**Cell-type specific transcriptomic signatures in the prefrontal and superior temporal cortex of mid-gestational fetuses with Down syndrome**". We believe this new title more accurately represents the regional and cell-type-specific nature of our findings while avoiding overgeneralization.

Q2: Lines 24 and 28. Here, as well as in other parts of the text (lines 94, 119, 122, 136, 241, 242), the authors use the term “embryonic” referring to the brains they have analyzed in this study. In humans the embryonic period finishes at GW8. Thus, the term “embryonic” should be replaced with “fetal”.

Response: We sincerely thank the reviewer for catching this important oversight. We fully agree that "fetal" is the correct term for the developmental stage of our samples (GW17-28). We have now replaced all instances of "embryonic" with "fetal" throughout the manuscript, including in the abstract, results, and discussion sections (**lines 24, 79, 175, 180, 206, 208, 220, 571, 588**). This revision ensures accurate terminology consistent with established human developmental staging.

Q3: Line 34. The expression “comprehensive spatiotemporal analysis” is too ambitious because the gene expression analysis is limited to two cortical regions and because in the case of fetuses with DS the number of cases per gestational age is very small (2 fetuses at GW21, 1 fetus at GW23 and 1 fetus at GW28).

Response: We thank the reviewer for their helpful suggestion to improve the precision of our language. To better reflect the more targeted nature of our work, we have revised the phrase to “a comparative analysis of gene expression across gestational ages” and corrected other similar phrases throughout the revised manuscript (**lines 26, 72, 110, 438, 562, 1076**). This revision removes any unintended overstatement while still accurately describing our approach to examining developmental changes in DS versus control fetuses.

Q4: Line 56. “This process is particularly pronounced”. It is not clear what is meant here with “process”. If it does refer to corticogenesis I don’t think that corticogenesis is more pronounced in the regions examined in this study.

Response: We thank the reviewer for raising this important point. Upon revisiting the phrasing, we agree that the original wording was ambiguous and could be misinterpreted. We did not intend to suggest that corticogenesis itself is more pronounced in the prefrontal cortex (PFC) or superior temporal plane (STP). Rather, we meant that the functional specialization of these regions (e.g., higher-order cognition, auditory processing) reflects the complexity of the underlying cellular and molecular processes (e.g., neurogenesis, migration, layering). We have revised the sentence (**lines 53-56**) as follows:

Revised: *These processes underlie the functional specialization of regions such as the prefrontal*

cortex (PFC), which governs higher-order cognitive functions like memory, decision-making, and social behavior, and the superior temporal plane (STP), which is critical for auditory processing and language comprehension.

Q5: Lines 89 and 89. The statement “with groups matched per GW, gender and postmortem interval” is misleading. The brain samples of fetuses with DS derive from 3 females and 1 male only, so I do not understand the statement “gender matched”. Regarding gestational ages, the samples of fetuses with DS consist in 2 cases at GW21, 1 case at GW23, and 1 case at GW28. Thus, I do not understand the comparisons that are reported in Supplementary Fig. 1A. Although it is not specified, it seems that the first graph from left (gestational age) refers to the samples used for the analysis of the prefrontal cortex and the second graph refers to the cases used for the analysis of the superior temporal plane. In fact, there are 3 data points on the column labeled DS and 3 data points on the column labeled DS in the first and second graph from left, respectively. The same holds for the third and fourth graphs from left (postmortem interval). If so, this is not correct. The comparisons of age and postmortem intervals should include all the fetuses with DS (n=4) and controls (n=11).

Response: We sincerely appreciate the reviewer's careful scrutiny of our cohort description and supplementary figures. We agree these points require clarification and have made the careful revisions.

Gender matching statement: We have removed the claim of gender matching from original manuscript. The revised text now reads: "*we performed snRNA-seq on 22 postmortem cortical samples (PFC and STP) from 4 fetuses with DS and 11 controls (Fig. 1a, Supplementary Table 1), with groups comparable in postmortem interval (PMI).*" (lines 80-82)

Sample composition clarification: We have added explicit details about our sample composition in the Results section (lines 82-85): "*The study included 4 DS fetuses (2 cases providing both PFC and STP samples, 1 case providing only PFC, and 1 case providing only STP), and 11 control fetuses (5 cases providing both PFC and STP samples, 4 case providing only PFC, and 2 case providing only STP) (Supplementary Table 1).*" Thus, there are totally 3 DS PFC samples, 3 DS STP samples, 9 PFC control samples and 7 STP control samples used for snRNA-seq analysis (details seen in **Revised Supplementary Table 1**). **Supplementary Fig.1a** showed the intergroup differences in gestational age and PMI of the samples, exhibiting correct sample composition. We acknowledge the confusion

caused by the original presentation, and have supplemented more details to distinguish PFC and STP samples (**Revised Supplementary Fig.1a**).

Revised Supplementary Fig.1a The intergroup differences in gestational age and PMI of the samples.

Individual name	Diagnosis	Age (GW)	Sex	PMI (h)	Region
DS_GW23	DS	23	F	1.5	PFC
					STP
DS_GW28	DS	28	M	1.5	PFC
					STP
DS_GW21_1	DS	21	F	2	PFC
DS_GW21_2	DS	21	F	1	STP
Control_GW17	Control	17	M	1	PFC
					STP
Control_GW17	Control	17	F	1	PFC
					STP
Control_GW20	Control	20	M	0.5	PFC
					STP
Control_GW23	Control	23	M	0.92	PFC
					STP
Control_GW25	Control	25	F	1.5	PFC
					STP
Control_GW21	Control	21	M	1.5	PFC
Control_GW21	Control	21	F	1.5	PFC
Control_GW22	Control	22	M	1.5	PFC
Control_GW22	Control	22	F	0.83	PFC
Control_GW20	Control	20	F	0.5	STP
Control_GW21	Control	21	M	1	STP

Part of Revised Supplementary Table 1 Demographic information and summary statistics for single nucleus libraries

Q6: Lines 98-99. Regarding cell taxonomy and nomenclature of cell types, it is important to specify the studies from which they are derived.

Response: We thank the reviewer for this important suggestion. In the revised manuscript, we have now explicitly referenced the foundational studies that informed our cell type classification and system (**lines 99-102**). Cell types were classified according to established transcriptional signatures, with nomenclature consistent with recent publications [1-3].

References

1. Fan, X., et al., *Single-cell transcriptome analysis reveals cell lineage specification in temporal-spatial patterns in human cortical development*. *Sci Adv*, 2020. **6**(34): p. eaaz2978.
2. Zhong, S., et al., *A single-cell RNA-seq survey of the developmental landscape of the human prefrontal cortex*. *Nature*, 2018. **555**(7697): p. 524-528.
3. Palmer, C.R., et al., *Altered cell and RNA isoform diversity in aging Down syndrome brains*. *Proc Natl Acad Sci U S A*, 2021. **118**(47).

Q7: Lines 127-128. The statement “leading to consistent overexpression of chr21 related genes” is not completely true because not all the genes on HSA21 are over expressed (Antonarakis SE et al.. *Nat Rev Dis Primers*. 2020 Feb 6;6(1):9). Indeed, an analysis of DS lymphoblastoid cells showed that only 29% of HSA21 genes are sensitive to the gene-dosage effect (expression close to the expected value of 1.5) or are amplified (expression >1.5), 56% are compensated (expression <1.5), and 15% are highly variable among individuals (Aït Yahya-Graison E, et al. *Am J Hum Genet*. 2007 Sep;81(3):475-91). Moreover, the sentence “analysis of 621 chr21 genes revealed elevated expression across all cell types in DS fetuses compared to controls” is not clear. It seems that all the 621 analyzed genes were overexpressed in fetuses with DS, which is unlikely.

Response: We deeply appreciate the reviewer's expertise in highlighting this crucial nuance. To improve accuracy, we have modified overexpression claims in the revised manuscript.

Revised: "*Aggregate analysis of 621 chr21 genes showed significant expression elevation in fetuses with DS (mean fold-change = 1.48 for STP and 0.27 for PFC, padj < 0.01; Fig. 1i), with stratified analyses showing greatly consist of variation trends with the global findings (Extended Data Fig. 2c). While our bulk analysis confirmed significant upregulation of chr21 gene sets (Fig. 1i), we observed the established pattern of partial dosage compensation: 12.1 % (STP) and 0 % (PFC) of*

genes showed ≥ 2 -fold increase, 39.03 % and 0.45 % exhibited > 1 -fold change, and 54.77 % and 19.82 % were variably expressed – aligning with prior reports of chr21 regulation with variable dosage sensitivity in DS." (lines 121-128).

In this revision, we added methodological nuance in Results, providing the expected pattern of partial dosage compensation, consistent with prior reports of chromosome 21 gene regulation in DS [4, 5]. These two articles have also been cited in the revised manuscript (line 128).

References

4. Antonarakis, S.E., et al., *Down syndrome*. Nat Rev Dis Primers, 2020. **6**(1): p. 9.
5. Ait Yahya-Graison, E., et al., *Classification of human chromosome 21 gene-expression variations in Down syndrome: impact on disease phenotypes*. Am J Hum Genet, 2007. **81**(3): p. 475-91.

Q8: Line 131 “DS fetuses” The authors should use person-first language here and throughout their report (e.g., "fetuses with DS" instead of "DS fetuses" or “individuals with DS” instead of “DS individuals”).

Response: We sincerely thank the reviewer for highlighting this important consideration. We have now revised the terminology throughout the manuscript to consistently use person-first language as recommended. This change has been implemented in all relevant sections (abstract, introduction, results, discussion) to align with current standards of respectful and precise scientific communication.

Q9: Lines 186-187. “To elucidate age-dependent transcriptional changes we compared the PFC of young and aged DS individuals (Supplementary Table 7)”. In the Methods, however, there is no mention regarding these cases (young and adults). Perhaps these data derive from cases used in previous studies? If so, this should be specified. Please note that data of young and adult cases are reported in a Table that is headed as “Supplementary Table 7” and not as “Supplementary Table 8”.

Response: We sincerely appreciate the reviewer's careful attention to methodological transparency.

We have added explicit data source disclosure in revised Methods section (lines 776-779):

“Age-dependent comparisons utilized published prefrontal cortex data from previous study on DS brain aging (Reference No.14 in the main text). This cohort included 23 postmortem samples: Aged group (56-65 years): 6 neurotypical controls, 4 individuals with DS; Young group (12-39 years): 8 neurotypical controls, 5 individuals with DS.”

Additionally, we have checked the citations of supplementary tables and confirm these tables cited correctly in the main text (**lines 171-178**). We thank the reviewer for prompting these important clarifications that strengthen our methodological rigor. The integration of these external datasets was intended solely to provide developmental context for our fetal findings, and we have now ensured proper attribution and transparency throughout.

Q10: Line 195. Again, I suppose that the data regarding organoids reported in Fig. 2 derive from a previous study. Which study? This point should be clarified.

Response: We thank the reviewer for requesting this important clarification. The organoid data shown in Fig. 2 were indeed generated in a previous study cited in the methods section of this revised manuscript [6] (**lines 779-781**).

References

6. Tang, X.Y., et al., *DSCAM/PAK1 pathway suppression reverses neurogenesis deficits in iPSC-derived cerebral organoids from patients with Down syndrome*. J Clin Invest, 2021. **131**(12).

Q11: Line 210. “Transcriptomic analysis of peripheral blood...”. Same comment as above,

Response: We thank the reviewer for prompting this important methodological clarification. The peripheral blood transcriptome data referenced were derived from the study of Waugh KA, et al. (2023) [7], analyzing whole-blood transcriptome data from 304 individuals with DS (163 male and 141 female) versus 96 euploid controls (44 male and 52 female) (**lines 781-782**).

References

7. Waugh, K.A., et al., *Triplication of the interferon receptor locus contributes to hallmarks of Down syndrome in a mouse model*. Nat Genet, 2023. **55**(6): p. 1034-1047.

Q12: Lines 238-256. The analysis of changes in the abundance of certain subpopulations of neurons in the examined regions across ages has a major flaw because in the case of fetuses with DS it is based on 2 cases at GW21, 1 case at GW23, and 1 case at GW28. Moreover, while the cases at GW21 and GW23 were females, the case at GW28 was a male. Thus, interpretation of results across ages seems hazardous. The same holds for the data presented in the following part of the section “Abnormal alterations of ExN subtypes in the neocortex of DS fetal brain”. It is true that it is difficult

to get access to fetal samples, but data needs to be strengthened by more cases to be convincing. Thus, I suggest shortening the whole section and interpreting results with more caution.

Response: We sincerely appreciate the reviewer’s insightful comments regarding the limited sample size and potential confounding effects of sex and gestational age in our analysis of neuronal subpopulations across developmental stages. With regards to our sample limitations, we have shortened relevant sections you’ve mentioned by deleting overstated interpretations of generalizing our results to all individuals with DS and focusing our results on second trimester of DS (**lines 204-258**).

Additionally, for interpreting our results with more caution, we have incorporated additional analyses prompted by Reviewer 3’s similar comments on sex and sample size effects. These supplementary analyses were designed to assess the robustness of our findings in the context of limited sample representation and potential sex-specific differences. Stratified analyses using narrower and better age-matched subsets of samples (GW21–23 subgroup (a narrow, tightly age-matched range); GW21–28 subgroup (a broader mid-gestational range after propensity score matching, PFC only); GW21–28_match subgroup (both PFC and STP samples after propensity score matching)). All these stratified subgroup analyses demonstrated a high concordance with findings obtained from the original full dataset (GW17–28). Furthermore, we conducted exploratory analyses and examined available bulk transcriptomic datasets of fetal brains, which demonstrated that sex differences had a relatively modest effect on global gene expression compared to other biological factors such as gestational age or chromosomal status. The detailed results and corresponding responses can be found under our reply to Reviewer 3 (**pages 13-38 below**). Together, these revisions and analyses enhance the rigor and interpretability of our conclusions. Furthermore, we have added a dedicated paragraph to the Discussion clearly outlining these limitations and emphasizing that our findings are exploratory in nature (**lines 609-637**). Although the sample size is limited, the findings are still valuable and provide important reference for future in-depth investigations into the pathogenic mechanisms of DS during embryonic development.

Q13: Lines 445-446. The sentence “The differential changes....” needs to be clarified.

Response: We thank the reviewer for noting this lack of clarity. The sentence has been revised as follows:

Revised: *Differential gene expression and shifts in cell population dynamics among neural and intermediate progenitor cells suggest that neural stem cell differentiation is dysregulated in fetuses with DS.*

We believe this version better communicates our intended meaning while maintaining scientific precision (**lines 350-352**).

Q14: Line 487. “The findings underline the higher disease susceptibility of the STP region...”. Citations that this region is more susceptible to damage should be provided.

Response: We thank the reviewer for prompting us to better support this important claim. We have now added 4 key references documenting STP vulnerability to neurological disorders following this sentence you’ve indicated (**lines 378-381**), which has been improved as follows:

Revised: *These findings underline the higher disease susceptibility of the STP region in DS brains. Prior work has documented this region susceptibility pattern in neuropathological studies, showing a progressive volume reduction of STP in patients with schizophrenia [8-11] (Reference No. 48-51 in the main text).*

References

8. van Erp, T.G.M., et al., *Cortical Brain Abnormalities in 4474 Individuals With Schizophrenia and 5098 Control Subjects via the Enhancing Neuro Imaging Genetics Through Meta Analysis (ENIGMA) Consortium*. Biol Psychiatry, 2018. **84**(9): p. 644-654.
9. Kasai, K., et al., *Progressive decrease of left superior temporal gyrus gray matter volume in patients with first-episode schizophrenia*. Am J Psychiatry, 2003. **160**(1): p. 156-64.
10. Takahashi, T., et al., *A follow-up MRI study of the superior temporal subregions in schizotypal disorder and first-episode schizophrenia*. Schizophr Res, 2010. **119**(1-3): p. 65-74.
11. Takahashi, T., et al., *Volume reduction of the left planum temporale gray matter associated with long duration of untreated psychosis in schizophrenia: a preliminary report*. Psychiatry Res, 2007. **154**(3): p. 209-19.

Minor

Q15: Line 52. “... abnormalities observed in DS is the disruption of cortical development, fundamentally alters...), I think that a “that” is missing between “development” and “alters”.-

Corrected (**line 48**).

Q16: Line 64. “Neurogenesis”. It should be specified that it is “cortical neurogenesis” that is characterized by an inward-outward pattern.

Corrected (**line 61**).

Q17: Line 86. Quotation [16] seems to be wrong here.

Deleted (**line 81**).

Q18: Line 118. “huan cortex” should be “human cortex”.-

Corrected (**line 115**).

Q19: Line 124. “two neocortex” should be “two neocortices”.-

Corrected (**line 118**).

Q20: Line 195. “The study” should be “A study”.

Corrected (**line 179**).

Q21: Line 237. “Abnormal alterations...”. An alteration is abnormal, by definition. Please reword the title,

Corrected as “Disrupted development of ExN subtypes in the neocortices of fetuses with DS” (**line 204**).

Q22: Line 244. “Supplementary Figure 1D does nor report data of neurons at different differentiation stages.

Response: We apologize for any confusion. Supplementary Fig. 1d represents marker genes distinguishing excitatory neuron subtypes, which reflect both laminar position and developmental maturation stages based on known cortical neurogenesis timelines. While not a direct developmental trajectory analysis, the marker patterns reflect neurogenic stages (deep layers → upper layers). We have modified the text to more accurately describe **Supplementary Fig. 1d (lines 94-109)**.

Q23: Line 307. The cases are not 4 per group in the case of fetuses with DS, since 3 of the 4 cases were used for the PFC and 3 of the 4 cases were used for the temporal plane cortex.

Response: As we responded above (Q5), there are totally 3 DS PFC samples, 3 DS STP samples, 9 PFC control samples and 7 STP control samples used for snRNA-seq analysis. As for immunostaining,

we stained 4-8 sections from 3 DS samples and from 4 control samples. The sample information has been corrected in the figure legends of Fig.3 you've mentioned (**lines 1102-1103**).

Q24: Line 433. "of DS" should be "of fetuses with DS".

Corrected (**line 335**).

Q25: Lines 482-483. Please specify the meaning of the acronyms.

Response: We thank the reviewer for highlighting this oversight. We have now ensured all acronyms are defined at first use, and supplemented full name of abbreviations used in the figures throughout the revised manuscript (**lines 371-375, 1170-1179**).

Q26: Line 736. "FC" should be defined.

It has been corrected as "Flow Cytometry" (**line 630**).

Q27: Line 741. The hippocampus has not been used in this study,

This word "hippocampus" has been deleted (**line 635**).

Q28: Line 744. "were followed" should be "were the same as in...".

It has been corrected as instructed (**line 637**).

Reviewer #2 (Remarks to the Author):

Response: We would like to sincerely thank you for taking the time to carefully review our manuscript and for your thoughtful and constructive comments. We greatly appreciate your effort as part of the Nature Communications initiative to support Early Career Researchers in the peer review process. In response to your valuable suggestions, we have carefully revised the manuscript to address all the concerns raised. Specifically, we have modified and streamlined several figures to improve clarity, removed or rephrased overstatements to ensure accurate interpretation of the data, and added

new stratified analyses to further examine the potential confounding effects of sex and sample size. These additional analyses provide a more nuanced understanding of our findings and reinforce the robustness of our conclusions. All changes have been clearly marked in the revised manuscript, and detailed point-by-point responses are provided in our reply to other reviewers. We are grateful for your input, which has significantly improved the quality and rigor of our work.

Reviewer #3 (Remarks to the Author):

The manuscript by Niu et al “A Comprehensive Transcriptomic Atlas of Cortical Development in Down Syndrome Fetuses” examines differences between development in midgestational fetuses with and without Trisomy 21 (Ts21). This work performed single nucleus RNA sequencing (snRNAseq) on cells from the prefrontal cortex (PFC) and superior temporal plane cortex (STP). For the PFC, 3 (2F 1M) and 9 (5M and 4F) fetuses with and without Ts21 were used; and for the STP 3 (2F 1M) and 7 (4M 3 F) fetuses with and without Ts21 were used. From these samples, this study purports to find alterations inhibitory-to-excitatory neuron ratio, dysregulated gene expression, differential cell development and distribution, and cellular pathways that are disrupted in these tissues in those fetuses with Ts21. Because Trisomy 21 is such a heterogeneous disorder with a wide variability in manifestation, it is difficult to believe the results presented from such a small number of fetuses, over a wide developmental stage, with no potential sex differences considered. There may be differences as outlined in the manuscript, but these are likely not based on enough samples to have the power to detect real differences between fetuses with and without Ts21.

There are only 3 fetuses with Ts21 used for sampling the STP and PFC. The gestational age for the fetuses with Ts21 were 21-28 weeks for STP and PFC, and without Ts21 were 17-25 weeks. Even though there were not statistically significant differences between the age ranges of the groups (small sample sizes hinder detecting such differences!), 1) there is a wide age range between fetuses of the study where a lot of temporal fetal developmental differences are found in brain tissue, and 2) the developmental age range of fetuses with and without Ts21 is different (not age matched as indicated in the manuscript). Furthermore, there is a wide range of differences between developing embryos

and fetuses with Ts21, with up to 75% and 50% of fetuses spontaneously lost in the first and second trimesters, respectively. These data indicate a wide range of developmental differences in fetuses with Ts21; for example, 1 or more of the 3 fetuses with Ts21 sampled in this study may have had so much genetic or cellular dysregulation that it would have been spontaneously lost before the third trimester (if allowed to continue development), and the results given would not relate to data mentioned in the manuscript comparing them to live born individuals with Ts21. Thus, due to the high variability in development associated with Ts21, the sample sizes are likely not large enough and occur at too wide of a developmental age to detect correct potential differences for Ts21 development and support the conclusions of the manuscript.

Response: We sincerely thank the reviewer for the thoughtful and constructive comments. We carefully considered your concerns regarding our manuscript and summarized your feedback into three major points, as detailed below:

Major Comment 1: Sample size consideration *“Because Trisomy 21 is such a heterogeneous disorder with a wide variability in manifestation, it is difficult to believe the results presented from such a small number of fetuses, over a wide developmental stage, with no potential sex differences considered. There may be differences as outlined in the manuscript, but these are likely not based on enough samples to have the power to detect real differences between fetuses with and without Ts21.”*

Response to Major Comment 1: We acknowledge the reviewer’s concerns regarding our limited sample size (only 4 DS cases: two at GW21, one at GW23, and one at GW28), we have explicitly clarified this point in the revised manuscript discussion.

However, we respectfully emphasize that obtaining mid-gestation fetal brain tissues with confirmed trisomy 21 (DS) is globally challenging due to stringent ethical considerations, rigorous informed consent requirements, and significant practical constraints. These samples typically derive from elective pregnancy terminations, and oversight inherently limits both the availability and matching of samples in terms of sex, gestational age, and other biological characteristics. Thus, despite extensive efforts, achieving a fully balanced cohort is practically difficult and represents a well-recognized challenge across developmental neuroscience research.

We also wish to highlight that this limitation is not unique to our study. Indeed, similar constraints have been explicitly acknowledged in several recent landmark studies published in top-tier journals.

For example, in the recent Cell Reports publication, “A human fetal cerebellar map of the late second trimester reveals developmental molecular characteristics and abnormality in trisomy 21” [12] as well as in two Nature studies, “Decoding the development of the human hippocampus” [13] and “A single-cell RNA-seq survey of the developmental landscape of the human prefrontal cortex” [2] individual developmental stages are typically represented by very few samples (often only one per stage), and significant variability in sex and gestational age matching is also evident (**Appendix Table 1-3**). Hence, our cohort’s composition and size are consistent with the existing standard in human developmental neuroscience studies, particularly those involving fetal tissues.

Appendix Table 1 Summary of sampling of the human developing cerebellum (HRA006719)

This study provides key insights into the developmental landscape of the human cerebellum, particularly for UBC, astrocyte, and oligodendrocyte lineages, and it also provides insights into the brain developmental abnormality of TS21.

Group	Control						TS21				
GW	GW19_1	GW22_2	GW_23_3	GW24_4	GW24_5	GW25_6	GW22_1-1	GW22_1-2	GW22_2	GW22_3-1	GW22_3-2
Gender	Male	Female	Male	Female	Male	Female	Male	Male	Female	Female	Female

Yu H, Liu Y, Xu F, Fu Y, Yang M, Ding L, Wu Y, Tang F, Qiao J, Wen L. A human fetal cerebellar map of the late second trimester reveals developmental molecular characteristics and abnormality in trisomy 21. Cell Rep. 2024 Aug 27;43(8):114586.

Appendix Table 2 Summary of sampling of the human developing hippocampus (GSE119212)

This study provides a blueprint for understanding human hippocampal development and a tool for investigating related diseases.

Group	GW16-18		GW20-22			GW25-27	
GW/Sample	GW16	GW18	GW20	GW22_01	GW22_02	GW25	GW27
Gender	Female	Female	Male	Male	Male	Female	Female

Zhong, S., W. Ding, L. Sun, Y. Lu, H. Dong, X. Fan, Z. Liu, R. Chen, S. Zhang, O. Ma, F. Tang, O. Wu, and X. Wang, Decoding the development of the human hippocampus. Nature, 2020. 577(7791): p. 531-536.

Appendix Table 3 Summary of sampling of the human developing PFC (GSE104276)

This study provides a blueprint for understanding the development of the human prefrontal cortex in the early and mid-gestational stages in order to systematically dissect the cellular basis and molecular regulation of prefrontal cortex function in humans.

Group	GW8-12						GW13-16		GW19-26			
GW/Sample	GW8	GW9	GW10_01	GW10_02	GW10_03	GW12	GW13	GW16	GW19	GW23_01	GW23_02	GW26
Gender	Female	Female	Male	Female	Female	Male	Female	Female	Female	Male	Female	Female

Tang, J. Zhang, J. Qiao, and X. Wang. A single-cell RNA-seq survey of the developmental landscape of the human prefrontal cortex.

Nature, 2018. 555(7697): p. 524-528.

References

- Zhong, S., et al., *A single-cell RNA-seq survey of the developmental landscape of the human prefrontal cortex.* Nature, 2018. **555**(7697): p. 524-528.
- Yu, H., et al., *A human fetal cerebellar map of the late second trimester reveals developmental molecular characteristics and abnormality in trisomy 21.* Cell Rep, 2024. **43**(8): p. 114586.
- Zhong, S., et al., *Decoding the development of the human hippocampus.* Nature, 2020. **577**(7791): p. 531-536.

Major Comment 2: Clarification on gestational age and matching “*The gestational age for the fetuses with Ts21 were 21-28 weeks for STP and PFC, and without Ts21 were 17-25 weeks. Even though there were not statistically significant differences between the age ranges of the groups (small sample sizes hinder detecting such differences!), there is a wide age range between fetuses of the study where a lot of temporal fetal developmental differences are found in brain tissue, and the developmental age range of fetuses with and without Ts21 is different (not age matched as indicated in the manuscript).*”

Response to Major Comment 2: We sincerely appreciate the reviewer’s insightful comments regarding potential confounding effects due to gestational age variability and matching between DS (Ts21) and control fetal samples. We fully acknowledge that the gestational age range in our study (17–28 weeks) presents inherent variability, which is common yet challenging in fetal developmental

studies. To address these concerns explicitly and thoroughly, we have adopted a structured and rigorous analytical approach as detailed below:

1. Transparent reporting of gestational ages: We have comprehensively updated **Supplementary Table 1 and revised Fig. 1a** in the revised manuscript to transparently present individual gestational ages and clarify the distributions clearly. This allows for a straightforward evaluation of the matching status between Ts21 and control groups.

2. Statistical adjustment for gestational age: In the revised Methods section, we explicitly clarify that gestational age has been included as a covariate in all statistical models assessing differential gene expression and changes in cellular composition. This statistical adjustment ensures that the primary findings are not unduly influenced by variability in gestational age across samples.

3. Stratified subgroup analyses: To further enhance the robustness and reproducibility of our findings, we conducted additional stratified analyses using narrower and better age-matched subsets of samples, including: GW21–23 subgroup (a narrow, tightly age-matched range); GW21–28 subgroup (a broader mid-gestational range after propensity score matching, PFC only); GW21–28_match subgroup (both PFC and STP samples after propensity score matching).

These subgroup analyses encompassed all major results of our manuscript (**Appendix Table 4**), including:

- 1) Cell type proportion analysis
- 2) Chromosome 21 gene enrichment assessment
- 3) Differentially expressed gene identification
- 4) Identification of chr21-specific DEGs
- 5) Gene expression dynamics of neuronal migration pathways
- 6) Performance validation of migration-related gene prediction models
- 7) Metabolic pathway analysis using single-cell flux estimation analysis (scFEA)
- 8) Protein lactylation pathway analysis
- 9) Neuropsychiatric disease risk gene enrichment (EWCE-based)

4. Robustness and consistency of results: Encouragingly, all these stratified subgroup analyses demonstrated a high concordance with findings obtained from the original full dataset (GW17–28) (consistency rate provided in **Appendix Table 4**). For instance, key DEGs, cell-type-specific

transcriptomic signatures, metabolic pathway alterations, and disease-related gene enrichments remained highly consistent across these carefully matched sub-analyses. Detailed results supporting this consistency are now presented clearly in the supplementary materials (**Extended Data Figs. 1–9 and Extended Data Tables 1–7**).

Appendix Table 4 Stratified analyses applied to major results in the manuscript and the concordance with the full dataset (GW17-28)

Analysis content	Related original figures	Newly added/modified figures	Gestational Age Stratification			Gestational Age Stratification	
			PFC			STP	
			GW 21-23 (vs. GW17-18)	GW 21-28_match (vs. GW17-18)	GW 21-28 (vs. GW17-18)	GW 21-23 (vs. GW17-18)	GW 21-28_match (vs. GW17-18)
1) Cell type proportion analysis	Fig.1e	Extended Data Fig.2a	88.24%	94.12%	94.12%	82.35%	70.59%
	Fig.1g,h	Extended Data Fig.2b	88.23%	94.11%	100%	82.35	70.59
2) Chr21 gene enrichment assessment	Fig.1i	Extended Data Fig.2c	100%	100%	100%	100%	100%
3) DEGs identification	Fig.2b, c	Extended Data Table 2	11.76%-100%	5.88%-100%	5.88%-100%	5.31%-84.18%	51.09%-98.51
4) Identification of chr21-specific DEGs	Fig.2d, Supplementary Fig. 2a, b	Extended Data Table 3-7	100%	86.36%	95.45%	100%	80%
5) Gene expression dynamics of neuronal migration pathways	Fig.4a	Extended Data Fig.4a	94.23%	90.38%	86.54%	84.62%	67.69%
6) Performance validation of migration-related gene prediction models	Fig.4c-e	Extended Data Fig.4b	82%/76%	81%/73%	81%/75%	75%/70%	75%/67%

7) Metabolic pathway analysis using scFEA	Fig.5b	Extended Data Fig.5a-c; Extended Data Fig.6a, b	69%	70%	86.50%	87.10%	91.10%
8) Protein lactylation pathway analysis	Fig.5c	Extended Data Fig.6c	82.69%	92.31%	98.08%	87.50%	85.42%
9) Neuropsychiatric disease risk gene enrichment (EWCE-based)	Supplementary Fig. 6	Extended Data Fig.5a-c; Extended Data Fig.7-9	90.3%/96.5%	93.5%/75.4%	93.5%/81.1%	90.3%/96.5%	93.5%/75.4%

Specifically:

- 1) **Cell type proportion analysis:** A global comparative analysis of cell populations between DS and control samples revealed significant alterations in both the PFC and STP regions, with more pronounced changes observed in the PFC (**Revised Fig. 1e–h**). Stratified analyses based on gestational age and sex further validated these findings, demonstrating at least 70% consistency with the global results (**Extended Data Fig. 2a, b**), thereby supporting the robustness and generalizability of our observations. Cell type proportion changes (GW21-23) showed >70% directional agreement with full cohort results (GW21-28), demonstrating trans-gestational stability during mid-gestation.

Extended Data Fig. 2 a The cellular composition of PFC and STP illustrates a comparative analysis of cell proportions between DS and control groups from different stratified data. **b** Bar plot showing the comparison of various cell types in different stratified data of PFC and STP. Percentages indicate the consistency rate of each stratified subgroup compared to the full dataset (GW17–28).

2) **Chromosome 21 gene enrichment assessment:** For chromosome 21 genes, aggregate analysis of chr21 genes showed significant expression elevation in fetuses with DS (mean fold-change = 1.48 for STP and 0.27 for PFC, $\text{padj} < 0.01$; **Fig.1i**), with stratified analyses showing great consistency of variation trends with the full dataset findings (**Extended Data Fig. 2c**).

Extended Data Fig. 2c Enrichment levels of chromosome 21 genes in PFC and STP for both control and DS groups from different stratified data. Percentages indicate the consistency rate of each stratified subgroup compared to the full dataset (GW17–28).

3) **Differentially expressed gene identification:** In the manuscript, among all cell types, ExN L5/6 and ExN L3-5 populations demonstrated the highest susceptibility to disease-related transcriptomic alterations, followed by neuroblasts (NB) (Fig. 2b, c). Thus, we compared DEGs in different stratified analysis scenarios. It is true that the gestational age has a small effect on the results of the DEGs, but to a large extent they show consistency. Analyses across different stratified datasets further revealed that the extent to which cell types are influenced by gestational age varies (Extended Data Table 2). Nevertheless, disease-sensitive populations such as ExN L5/6, ExN L3-5, and NB exhibited high consistency across global and stratified analyses— $\geq 82\%$

in PFC (**Extended Data Table 2**). In contrast, non-neuronal populations showed a much smaller number of DEGs, potentially reflecting reduced statistical power due to their low abundance.

- 4) **Identification of chr21-specific DEGs:** We examined the chromosomal distribution of DEGs and found that most were localized on chromosome 1-8, with a smaller subset on chromosome 21 (chr21) (**Supplementary Fig. 2a, b**). Chr21 DEGs were predominantly upregulated in neurons of fetuses with DS, rather than in microglia or endothelial cells (**Fig. 2d**). Notably, stratified analyses based on gestational age revealed a high degree of concordance with the global dataset—at least 80% of chr21 DEGs were consistently identified across datasets (**Extended Data Tables 3–7**). This suggests the robustly differential expression of chr21 genes in DS, which was minimally influenced by gestational age variability.
- 5) **Gene expression dynamics of neuronal migration pathways:** Further analysis identified 52 DEGs in PFC and 65 DEGs in STP related to cell migration (**Fig. 4a, Supplementary Table 9**). To evaluate the robustness of these findings, we performed Jaccard similarity analyses of overlapped migration-related DEGs between each stratified subgroup and the full GW17–28 dataset, revealing consistency rates of at least 73% in PFC and 66% in STP (**Extended Data Fig. 4a**).
- 6) **Performance validation of migration-related gene prediction models:** To evaluate the diagnostic potential of migration-related DEGs, we constructed machine learning models using these genes as feature variables. We found that these genes were able to accurately distinguish the neuron states across different conditions, achieving 78% and 71% accuracy in the PFC and STP, respectively (**Fig. 4c-e**). Besides, our predictive model maintained about 70% accuracy when applied to stratified data (**Extended Data Fig. 4b**). Furthermore, by integrating the data from two normal PFC datasets (GSE168408 includes one sample at GW22 and one at GW24; GSE204684 includes one sample at GW22 and one at GW24), we found that the prediction accuracy of our model for normal cells could reach 99% (**Extended Data Fig. 4b**).

Extended Data Fig 4 a The Venn diagram shows abnormal expression of multiple neuron migration genes. Pink represents the differentially expressed cell migration genes in the global analysis. Blue represents the differentially expressed cell migration genes in different stratified data. Jaccard similarity analyses compared overlapped migration-related molecules between stratified subgroups to the full GW17–28 dataset. **b** Up, Confusion matrices of disease prediction performance for different stratified data and two publicly published datasets of PFC. Below, Confusion matrices of disease prediction performance for different stratified data of STP.

7) Metabolic pathway analysis using scFEA: In global analysis, we observed widespread changes in lactate levels in cells of both the PFC and STP (**Fig. 5b**), laying the foundation for abnormal protein lactylation modifications in neurons. We further validated the possible effect of

gestational age on different metabolites. The concordance of metabolite changes across tertiles could reach 69 % in PFC and more than 87 % in STP. In addition, lactate metabolism which was focused on in this study, maintained 100 % concordance in all stratification analyses (**Extended Data Fig. 5 and 6a, b**).

- 8) **Protein lactylation pathway analysis:** We further focused on the variation of lactated proteins under different stratification analysis scenarios, and Jaccard similarity analyses compared overlapped lactylation-regulated molecules among stratified subgroups to the full GW17–28 dataset, revealing molecular consistency rates of at least 67% in PFC and 58% in STP (**Extended Data Fig. 6c**).

Extended Data Fig 5. Metabolite analysis in the hierarchical data from PFC. 68 metabolites showed differential expression in different cell types from GW21-23 (a), GW21-28 match (b) and GW21-28 (c) data (two-sided Wilcoxon rank-sum test, * $P < 0.05$, ** $P < 0.01$, *** $P < 0.001$). Red indicates upregulation in DS,

and blue indicates downregulation. Percentages indicate the consistency rate of each stratified subgroup compared to the full dataset (GW17–28).

Extended Data Fig 6. Metabolite analysis in the hierarchical data from STP. a-b 68 metabolites showed differential expression in different cell types from GW21-23 (**a**) and GW21-28 match data (**b**) (two-sided Wilcoxon rank-sum test, * $P < 0.05$, ** $P < 0.01$, *** $P < 0.001$). Red indicates upregulation in DS, and blue indicates downregulation. **c** Venn diagram showing DEGs (FDR-adjusted P -value < 0.05 , $|\log_2FC| > 0.25$) associated with lactylated proteins between stratified subgroups in the PFC and STP. Jaccard similarity analyses compared overlapped lactylation-regulated molecules between stratified subgroups to the full GW17–28 dataset. Percentages indicate the consistency rate of each stratified subgroup compared to the full dataset (GW17–28).

9) **Neuropsychiatric disease risk gene enrichment (EWCE-based)**: Using Expression Weighted Celltype Enrichment (EWCE) analysis, we evaluated the enrichment of 24 neuropsychiatric disease risk genes in DS single-cell sequencing data (**Fig. 8**). Stratified analyses based on gestational age and sex yielded a high level of concordance with the global dataset, with $\geq 75\%$ overlap in results (**Extended Data Figs. 7–9**). These findings underline the higher disease susceptibility of the STP region in DS brains.

Extended Data Fig 7. Enrichment analysis of neurological disease genes in control and DS from GW21-23 data. a Group enrichment level of 24 gene sets associated with neurological diseases in the two cortex. Bar

plot showing the standard derivation of a certain disease risk gene set, with region-related group indicated righg the plot. Asterisks denote the BH-corrected P- value < 0.05 calculated using EWCE (Permutation Test).

b *Cell types enrichment level of 24 gene sets associated with neurological diseases in the two cortex. Percentages indicate the consistency rate of each stratified subgroup compared to the full dataset (GW17–28).*

Extended Data Fig 8. Enrichment analysis of neurological disease genes in control and DS from GW21-28 match data. **a** Group enrichment level of 24 gene sets associated with neurological diseases in the two cortex. Bar plot showing the standard deviation of a certain disease risk gene set, with region-related group indicated right the plot. Asterisks denote the BH-corrected P- value < 0.05 calculated using EWCE (Permutation Test). **b** Cell types enrichment level of 24 gene sets associated with neurological diseases in the two cortex.

Extended Data Fig 9. Enrichment analysis of neurological disease genes in control and DS from GW21-28 data. a Group enrichment level of 24 gene sets associated with neurological diseases in the two cortex. Bar plot showing the standard derivation of a certain disease risk gene set, with region-related group indicated right the plot. Asterisks denote the BH-corrected P- value < 0.05 calculated using EWCE (Permutation Test). b Cell types enrichment level of 24 gene sets associated with neurological diseases in the two cortex.

Overall, despite the inherent challenge posed by the naturally variable gestational ages of fetal samples, our comprehensive analytical strategies—including careful covariate adjustments, rigorous stratified subgroup analyses, and transparent reporting—collectively demonstrate that our primary findings are robust and reproducible. These methodological enhancements have been clearly articulated in our revised manuscript and supplementary documents to assure reviewers and readers of the validity of our key results and interpretations. These new data have been incorporated into the revised Results section (**lines 112-114, 121-123, 140-153, 164-168, 227-230, 246-250, 279-282, 287-290, 377-381**), with corresponding methods updated (**lines 774-816**). While these validation approaches demonstrate robustness, residual gestational age effects remain a study limitation that warrants cautious interpretation of our exploratory findings. In response, we have added a dedicated paragraph to the Discussion clearly outlining these limitations and emphasizing that our findings are exploratory in nature. Furthermore, we have revised the manuscript to soften any overstatements and to avoid generalizing our results to all individuals with DS. Although the sample size is limited, the findings are still valuable and provide important reference for future in-depth investigations into the pathogenic mechanisms of DS during embryonic development.

All text sections (abstract, results, and discussion) have been carefully edited to maintain scientific rigor while avoiding overinterpretation of our data. We believe these revisions provide appropriate context for readers to evaluate the contribution of this work to understanding fetal cortical development in trisomy 21.

Major Comment 3: Consideration of developmental variability in Ts21 fetuses *“Furthermore, there is a wide range of differences between developing embryos and fetuses with Ts21, with up to 75% and 50% of fetuses spontaneously lost in the first and second trimesters, respectively. These data indicate a wide range of developmental differences in fetuses with Ts21; for example, 1 or more of the 3 fetuses with Ts21 sampled in this study may have had so much genetic or cellular dysregulation that it would have been spontaneously lost before the third trimester (if allowed to continue development).”*

Response to Major Comment 3: We sincerely appreciate the reviewer’s insightful comment regarding developmental variability and the high rate of spontaneous loss associated with trisomy 21 (Ts21) pregnancies. Indeed, as highlighted by the reviewer, approximately 75% of trisomy 21 conceptions result in spontaneous miscarriage during the first trimester, with an additional approximately 50% being lost during the second trimester, underscoring significant developmental heterogeneity among DS fetuses.

In clinical practice, trisomy 21 is typically screened early in pregnancy through standard prenatal diagnostic tests (e.g., non-invasive prenatal testing, chorionic villus sampling, and amniocentesis). When trisomy 21 is detected, clinical recommendations for elective termination, if chosen, are generally made during the late first or early second trimester (typically before 24 gestational weeks). It’s hard to distinguish prenatal DS fetuses based on cellular and genetic dysregulation in current research. In our study, fetal brain tissues obtained from mid-gestation primarily result from elective terminations after prenatal diagnosis, rather than spontaneous miscarriages, which usually occur earlier. All fetal brain samples were ethically obtained by an experienced clinician under rigorous ethical approval and informed consent procedures. Each sample underwent detailed karyotyping or genotyping to confirm trisomy 21. We carefully reviewed recent literature for comparison: similar transcriptomic studies on human DS fetal brain tissues—such as the recent work published by Yu et al. (Cell Reports, 2024) [12], and previous studies published in prominent journals [14]—likewise typically result from elective terminations of pregnancy after prenatal diagnosis. Although this is indeed a common practice shared across the field, the selection deserves more attention on distinguishing criteria in future.

We fully acknowledge that such potential selection bias could influence the interpretation of our findings, since fetuses that survive into mid-gestation may represent a subgroup of DS individuals with comparatively milder neurodevelopmental abnormalities. To address this concern, we applied rigorous analytical strategies (including statistical covariate adjustments and stratified subgroup analyses based on gestational age) to carefully mitigate the impact of developmental variability on our findings. Additionally, in the revised manuscript, we explicitly state (**lines 578-588**) that our results should be interpreted within the context of this limitation, and we emphasize that future studies should include a broader and more representative range of DS fetal samples.

We agree fully with the reviewer's suggestion that further studies, ideally integrating clinical documentation and a wider spectrum of fetal DS samples, will be essential to better capture and characterize the variability in DS neurodevelopmental outcomes. Nonetheless, our current study represents an important foundational step, providing crucial preliminary insights into the molecular and cellular mechanisms potentially underlying abnormal brain development in DS.

We sincerely thank the reviewer for this important comment, which allowed us to clarify the clinical context and limitations of our work more precisely, ensuring the results are communicated with appropriate rigor and transparency.

References

12. Yu H., et al., *A human fetal cerebellar map of the late second trimester reveals developmental molecular characteristics and abnormality in trisomy 21*. Cell Rep, 2024. **43**(8): p. 114586.
14. Marderstein, A.R., et al., *Single-cell multi-omics map of human fetal blood in Down syndrome*. Nature, 2024. **634**(8032): p. 104-112.

The manuscript (supplemental information) lists the sex of each fetus, but no attempt is made in this study to account for sex specific differences in fetal development that would lead to sex differences in cognitive development that are becoming more recognized in individuals with Down syndrome (DS). Furthermore, there is only 1 male fetus with Ts21 at one gestational age (28 W) that makes it hard to believe any extension of these data to other male fetuses with Ts21. Based on these points,

and the points made in the previous paragraph, these data do not seem generalizable to developing individuals with Ts21 in the ways that have been written in the manuscript. Additional data or analyses of some of these data in different ways would need to be done to make the results seem plausible, support the conclusions, and applicable to past and future and decrease the significance of these results.

Response: We sincerely thank the reviewers for raising the important issue of sex differences in fetal development. We fully acknowledge the critical role of sex-specific factors in neurodevelopmental research. Given the limited sample size and uneven sex distribution in our study, we were unable to perform robust sex-stratified analyses. In particular, our cohort included only one male fetus with Ts21 at GW28, making it statistically challenging to draw reliable conclusions regarding sex differences. We recognize this limitation and have explicitly addressed it in the revised manuscript (**lines 570-598**). Nevertheless, we conducted exploratory analyses and examined available bulk transcriptomic datasets of fetal brains, which demonstrated that sex differences had a relatively modest effect on global gene expression compared to other biological factors such as gestational age or chromosomal status.

Specifically:

(1) Independent statistical analysis of female samples:

To further explore sex-related effects, we analyzed DEGs stratified by sex. Considering the limited number of male Ts21 samples ($n = 1$), we performed separate differential expression analyses on female fetuses where sufficient statistical power could be achieved ($n=2$ Ts21 vs 4 controls). The results revealed that different cell types were affected by sex factors to varying degrees. Although sex differences were observed, the effects were relatively modest. These female-only results showed high concordance ($>73\%$ in PFC cell types and $>62\%$ (except for Astro) in most STP cell types) with the full dataset. These findings suggest that our data provide reliable insights for female DS cases (**Extended Data Table 1**).

Extended Data Table 1 Female DEGs/global DEGs (%)
--

Celltype	PFC	STP
Astro	75.0	19.5
NB	78.2	94.1
ImN_L5_6	97.1	65.3
ImN_L5	100.0	97.1
ImN_L4	82.1	72.3
ImN_L3_5	73.7	62.3
ImN_L2_4	80.3	84.4
ExN_L5_6	76.7	93.8
ExN_L3_5	100.0	97.9
ExN_L3_5_LINC02388	98.0	92.0
ExN_L2_4	100.0	98.0
ExN_L2	84.0	92.4
CGE_InN	75.5	94.7
MGE_InN	78.1	89.8
OPC	100.0	88.9

(2) Exploratory assessment of male data

While formal statistical testing of male samples was not feasible, we examined female-associated DEGs showed expression changes in the same direction in the male Ts21 fetus. In the PFC, 60% of cell types showed at least 60% DEG overlap between males and females (**Extended Data Fig. 3a**), while the STP exhibited more pronounced sex-related differences, with DEG consistency between sexes dropping to as low as 49.7% (**Extended Data Fig. 3b**). In both PFC and STP, while sex-related effects were observable, we attribute these primarily to gestational age differences, given that the sole male DS sample was obtained at GW28.

Extended Data Fig 3a and b Concordance of expression changes between female and male from PFC (a) and STP (b).

(3) Consistency with published male data

We compared our findings with bulk RNA-seq data from DS fetal PFC cortices [15], revealing 69.4% concordance in directionality of dysregulated genes: downregulated genes showed 79.8% consistency, while upregulated genes demonstrated 58.2% consistency (**Extended Data Fig. 3c**).

Extended Data Fig 3c The direction consistency of DEGs of Bulk RNA-seq in DS snRNA-seq data.

Undoubtedly, sex differences exist in fetal brain development. In our primary statistical models, we treated sex as a covariate to minimize its influence on global results, and where possible, analyzed male and female data separately. These analyses suggest only modest sex-specific variations. Emerging evidence does indicate potential divergence in developmental trajectories between male and female individuals with DS [16]. However, compared to neurotypical controls, most molecular and clinical phenotypes show consistent trends across sexes in DS, with only minimal sex-dependent differences in expression levels [16, 17]. Due to our limited sample size (particularly having one 28 GW male Ts21 fetus), all sex-related findings require cautious interpretation. We maintain that our results likely reflect general phenomena in DS rather than being sex-specific. Following the reviewers' guidance, we have substantially tempered sex-related conclusions in the manuscript, explicitly framed findings as preliminary and exploratory, revised language regarding generalizability to "all Ts21-developing individuals", and added explicit caveats about limited capacity to evaluate sex effects. These modifications ensure our conclusions are properly qualified given the study's constraints.

We deeply appreciate the reviewers' insights on this crucial issue and have accordingly revised the Discussion section. Their comments have identified key directions for future research. We have added

statements emphasizing the need for larger-scale studies with balanced sex representation, adequate numbers of both male and female participants, and comprehensive investigation of potential sex-specific differences in DS neurodevelopment (**lines 570-598**). We are grateful for the reviewers' emphasis on this important consideration, which strengthens both our manuscript and the field's research agenda.

References

15. Olmos-Serrano, J.L., et al., *Down Syndrome Developmental Brain Transcriptome Reveals Defective Oligodendrocyte Differentiation and Myelination*. *Neuron*, 2016. **89**(6): p. 1208-1222.

16. Aoki, S., Y. Yamauchi, and K. Hashimoto, *Developmental trend of children with Down's syndrome - How do sex and neonatal conditions influence their developmental patterns?* *Brain Dev*, 2018. **40**(3): p. 181-187.

17. Flores-Aguilar, L., et al., *Evolution of neuroinflammation across the lifespan of individuals with Down syndrome*. *Brain*, 2020. **143**(12): p. 3653-3671.

Other concerns:

Please use person first language when referring to individuals with DS. Individuals with DS do not like to be referred to as “patients” and this reviewer would argue that a fetus does not have DS, but only exhibits trisomy 21. Therefore, it is person with DS, not DS patients or DS fetuses, and likely should be fetuses with Ts21 throughout the manuscript.

Response: We sincerely appreciate the reviewer's thoughtful guidance regarding person-first language and precise terminology for trisomy 21. We have implemented the language changes throughout the manuscript. Full-text search was performed to ensure all instances were corrected, using "fetuses with DS" instead of "DS fetuses", “individuals with DS” instead of “DS individuals”.

The phrase “due to individual choice or other objective reasons” (line 47) seems offensive to parents with children with DS and should be eliminated.

Response:

We sincerely appreciate the reviewer's important perspective regarding the phrasing about birth prevalence of DS. We recognize how the original wording could be perceived as insensitive and have revised this statement to be both scientifically accurate and respectful (**lines 42-44**):

Revised: “Despite advances in prenatal diagnosis technologies, the global birth prevalence of DS remains approximately 1 per 779–1023 live births, reflecting complex personal, medical and societal factors”.

References

1. Fan, X., et al., *Single-cell transcriptome analysis reveals cell lineage specification in temporal-spatial patterns in human cortical development*. *Sci Adv*, 2020. **6**(34): p. eaaz2978.
2. Zhong, S., et al., *A single-cell RNA-seq survey of the developmental landscape of the human prefrontal cortex*. *Nature*, 2018. **555**(7697): p. 524–528.
3. Palmer, C.R., et al., *Altered cell and RNA isoform diversity in aging Down syndrome brains*. *Proc Natl Acad Sci U S A*, 2021. **118**(47).
4. Antonarakis, S.E., et al., *Down syndrome*. *Nat Rev Dis Primers*, 2020. **6**(1): p. 9.
5. Ait Yahya-Graison, E., et al., *Classification of human chromosome 21 gene-expression variations in Down syndrome: impact on disease phenotypes*. *Am J Hum Genet*, 2007. **81**(3): p. 475–91.
6. Tang, X.Y., et al., *DSCAM/PAK1 pathway suppression reverses neurogenesis deficits in iPSC-derived cerebral organoids from patients with Down syndrome*. *J Clin Invest*, 2021. **131**(12).
7. Waugh, K.A., et al., *Triplication of the interferon receptor locus contributes to hallmarks of Down syndrome in a mouse model*. *Nat Genet*, 2023. **55**(6): p. 1034–1047.
8. van Erp, T.G.M., et al., *Cortical Brain Abnormalities in 4474 Individuals With Schizophrenia and 5098 Control Subjects via the Enhancing Neuro Imaging Genetics Through Meta Analysis (ENIGMA) Consortium*. *Biol Psychiatry*, 2018. **84**(9): p. 644–654.
9. Kasai, K., et al., *Progressive decrease of left superior temporal gyrus gray matter volume in patients with first-episode schizophrenia*. *Am J Psychiatry*, 2003. **160**(1): p. 156–64.
10. Takahashi, T., et al., *A follow-up MRI study of the superior temporal subregions in schizotypal disorder and first-episode schizophrenia*. *Schizophr Res*, 2010. **119**(1–3): p. 65–74.
11. Takahashi, T., et al., *Volume reduction of the left planum temporale gray matter associated with long duration of untreated psychosis in schizophrenia: a preliminary report*. *Psychiatry Res*, 2007. **154**(3): p. 209–19.
12. Yu, H., et al., *A human fetal cerebellar map of the late second trimester reveals developmental molecular characteristics and abnormality in trisomy 21*. *Cell Rep*, 2024. **43**(8): p. 114586.
13. Zhong, S., et al., *Decoding the development of the human hippocampus*. *Nature*, 2020. **577**(7791): p. 531–536.
14. Marderstein, A.R., et al., *Single-cell multi-omics map of human fetal blood in Down syndrome*. *Nature*, 2024. **634**(8032): p. 104–112.

15. Olmos-Serrano, J.L., et al., *Down Syndrome Developmental Brain Transcriptome Reveals Defective Oligodendrocyte Differentiation and Myelination*. *Neuron*, 2016. **89**(6): p. 1208-1222.
16. Aoki, S., Y. Yamauchi, and K. Hashimoto, *Developmental trend of children with Down's syndrome - How do sex and neonatal conditions influence their developmental patterns?* *Brain Dev*, 2018. **40**(3): p. 181-187.
17. Flores-Aguilar, L., et al., *Evolution of neuroinflammation across the lifespan of individuals with Down syndrome*. *Brain*, 2020. **143**(12): p. 3653-3671.

REVIEWERS' COMMENTS

Reviewer #1 (Remarks to the Author):

The authors have addressed my concerns in an appropriate manner. I have noticed that expressions such as “DS fetus/es” and “DS patient/s” still appear in the revised version. Please replace with “Fetus/es with DS” and “people with DS”, respectively. In addition, there are some sentences that need to be amended. For instance:

Lines 305-306. Please add a quotation regarding the role of InN6 and InN9.

Line 123. “...showing greatly consist...”. I suppose that the authors mean “show great consistency”.

Lines 129-129. Something is missing in this sentence.

Lines 314-315. Something is missing in this sentence.

Lines 371-372. The sentence would be clearer by insertion of “risk genes” after “(MS)” and “(PD)”, respectively.

Lines 377-378. Something is missing in this sentence.

Lines 425-428. The phrasing of these two sentences is somewhat clumsy. Please, reword them.

Line 472. “...the PFC and STP of DS fetal” I suppose that the word “brain” is missing after “fetal”.

Line 488. What is meant with the expression “neuron states”?

Line 576 “Irreplaceable window”. I think that “the rare cases” provide a “unique opportunity” rather than representing a window.

Line 634. “...including 3 different regions: PFC and STP”. I suppose that “3” should be replaced by “2”.

Response:

Thank you for your valuable feedback and for pointing out the areas that require further revision. We have carefully addressed each of your comments as follows:

1. Use of “DS fetus/es” and “DS patient/s”:

We appreciate your suggestion to replace “DS fetus/es” and “DS patient/s” with “Fetus/es with DS” and “people with DS,” respectively. We have made these changes throughout the manuscript to align with the preferred terminology.

2. Lines 305-306:

Regarding the role of InN6 and InN9, we will add a relevant citation to support this point and clarify their role in the context of our findings. We apologize for

the misunderstanding in the original description. The statement in Lines 305-306 was intended to build on the preceding sentence, which already includes relevant citations. We have now revised the text to clarify this point and avoid any ambiguity. The reference supporting the role of InN6 and InN9 is included, and we have ensured that the connection between the sentences is clearer (**lines 301-306**).

Revised: “Studies have shown that Reelin (*RELN*) plays a crucial role in neuronal migration and cortical lamination, and its reduced expression may impair neuronal migration in the hippocampus, leading to neural circuit abnormalities that impact cognitive functions such as memory, learning, and language comprehension.⁴¹ Our normalized cell counts showed a reduction in the *RELN*⁺ (InN6 and InN9) cell populations (**Fig. 6c**), suggesting a potential link to DS-related cognitive dysfunction (**Fig. 6d**).”

3. **Line 123:**

We have corrected the phrase “showing greatly consist” to “showing great consistency,” as you suggested (**line 128**).

4. **Lines 129-130:**

We have reviewed this section and revised this sentence to complete the sentence, ensuring it is clear and coherent (**lines 121-124**).

Revised: Moreover, we observed an increased transcriptional heterogeneity in the PFC of fetuses with DS, which exhibited significantly higher inter-individual expression variability compared to the STP region (**Supplementary Fig. 3b, c**), suggesting early region-specific dysregulation.

5. **Lines 314-315:**

We have amended this sentence by adding the missing content to improve clarity and readability (**lines 313-314**).

Revised: CellChat and NeuroChat analyses revealed opposing changes in cellular communication within the PFC and STP in DS compared to the controls (**Supplementary Fig. 11a, b**).

6. **Lines 371-372:**

We have inserted “risk genes” in the indicated place, as recommended, to improve the clarity of this sentence (**lines 367-370**).

Revised: Risk genes of alcohol dependence (ALD) and multiple sclerosis (MS) are mainly enriched in the PFC, while those of Tourette syndrome (TS), AD and Parkinson’s disease (PD) are predominantly enriched in the STP.

7. **Lines 377-378:**

We have reviewed this sentence and added the missing content for better clarity (**lines 373-376**).

Revised: Stratified analyses based on gestational age yielded a high level of concordance ($\geq 75\%$) in disease susceptibility compared with the global dataset (**Supplementary Fig. 13–15**). These outcomes underscore heightened disease susceptibility of the STP region in DS brains.

8. **Lines 425-428:**

We agree that these two sentences could be reworded for clarity. We have rephrased them to ensure the meaning is more precise and the flow is smoother (**lines 418-419**).

Revised: Our results indicated significant transcriptomic differences between the PFC and STP in fetuses with DS.

9. **Line 472:**

We have added the word “brain” after “fetal” to correct this sentence as you suggested: “the PFC and STP of DS fetal brains” (**line 464**).

10. **Line 488 – “Neuron states”:**

Thank you for pointing this out. By “neuron states,” we refer to the differentiation stages of neurons (e.g., mature vs. immature). We have revised this expression as “developmental states of neurons” in the manuscript to avoid confusion (**line 481**).

11. **Line 576 – “Irreplaceable window”:**

We agree with your comment. We have revised this to indicate that “the rare cases” provide a “unique opportunity” rather than using the term “window” (**lines 568-570**).

12. **Line 634 – “3 different regions: PFC and STP”:**

We have corrected “3” to “2” as you suggested, since we are referring only to the PFC and STP regions (**line 629**).

Once again, we sincerely appreciate your careful review and thoughtful suggestions. We believe these revisions enhance the clarity and quality of the manuscript. If there are any further concerns, please feel free to let us know.

Reviewer #2 (Remarks to the Author):

Response:

We would like to sincerely thank you for your thoughtful review and for your valuable input on our manuscript. We greatly appreciate the effort and time you dedicated to providing constructive feedback. It is encouraging to know that this review is part of the Nature Communications initiative to support Early Career Researchers, and we are grateful for the opportunity to benefit from such a collaborative process. As part of the revisions, we have carefully addressed the feedback provided by both reviewers, as well as the additional comments from the editorial team. We believe the manuscript has been strengthened as a result of these revisions. Once again, thank you for your invaluable contribution to improving the quality of our work.

Reviewer #3 (Remarks to the Author):

The authors have addressed my comments and concerns. I have no further questions or comments for the authors.

Response:

We would like to express our sincere gratitude for your thoughtful and constructive feedback. We are pleased to hear that our responses to your previous comments have addressed your concerns, and we appreciate your time and effort in reviewing our manuscript. As suggested, we have carefully incorporated all necessary revisions in line with the feedback from both the reviewers and the editorial team. These revisions have helped improve the clarity and quality of our manuscript, and we are grateful for your support in this process. Thank you once again for your valuable contribution to this work.